# FairT2I: Latent Variable Guidance for Training-Free Bias Mitigation with LLM-Assisted Bias Detection

## Abstract

Text-to-image models have transformed visual content creation, but their reliance on large uncurated web data can encode and amplify societal biases. We present *FairT2I*, a training-free, inference-time framework that leverages large language models to detect implicit bias dimensions in prompts and mitigate them during generation. FairT2I has three components. First, LLM-based bias detection identifies bias-relevant attributes implied by the prompt and makes them explicit for control. Second, attribute resampling generates bias-aware prompts by sampling these attributes from a user-specified target distribution, including uniform, statistics-based, or custom choices. Third, *latent variable guidance* provides a theoretically grounded guidance rule that decomposes the diffusion score into attribute-conditioned components and reweights them to match the target attribute distribution. This can be viewed as an attribute-level generalization of classifier-free guidance, which applies a single global guidance scale to strengthen conditioning without explicit control over individual attributes. Experiments with both uniform targets and real-world employment statistics show that FairT2I outperforms existing training-free bias mitigation methods. On Parti Prompts, FairT2I improves diversity without sacrificing image quality or prompt fidelity, achieving a better quality–diversity trade-off than classifier-free guidance.

## 1 Introduction

Text-to-image (T2I) models, such as Stable Diffusion (Esser et al., 2024; Podell et al., 2023), Imagen (Imagen-Team-Google et al., 2024), and Flux (Black Forest Labs, 2024), have transformed visual content creation by allowing users to generate high-quality images from natural language descriptions. Yet, despite their accessibility and creative potential, these models inherit and amplify the biases present in large-scale web datasets (Bavalatti et al., 2025; Naik & Nushi, 2023). When prompted with neutral descriptions, models often produce stereotypical representations such as depicting a nurse as female or a chief executive officer as male, thereby reinforcing societal inequalities (Wan et al., 2024; Mandal et al., 2024; Sandoval-Martin & Martínez-Sanzo, 2024; Zhang et al., 2024b; de Almeida & Rafael, 2024). The increasing reuse of synthetic images in training pipelines further intensifies these biases over time (Wyllie et al., 2024; Alemohammad et al., 2023).

Many algorithmic approaches have been proposed to mitigate this problem; however, they continue to face significant practical limitations from the user's perspective. Large-scale dataset curation is challenging to sustain over time (Bianchi et al., 2023), and most existing strategies (Kim et al., 2024; Zhang et al., 2023; Hirota et al., 2024) rely on computationally expensive fine-tuning or retraining for each target bias, often involving delicate hyperparameter tuning (Friedrich et al., 2023). In addition, these approaches typically depend on manually pre-defined attribute lists (Kim et al., 2024; Zhang et al., 2023; Friedrich et al., 2023), which not only constrains their adaptability to emerging or context-dependent biases but also limits the degree of user control and interpretability during deployment.

To overcome these constraints, we present *FairT2I*, a novel, theoretically grounded framework that enables bias-mitigated diffusion-based T2I process. Fig. 1 illustrates the overview. FairT2I combines the reasoning ability of large language models (LLMs) with a flexible bias-aware image generation process. FairT2I operates

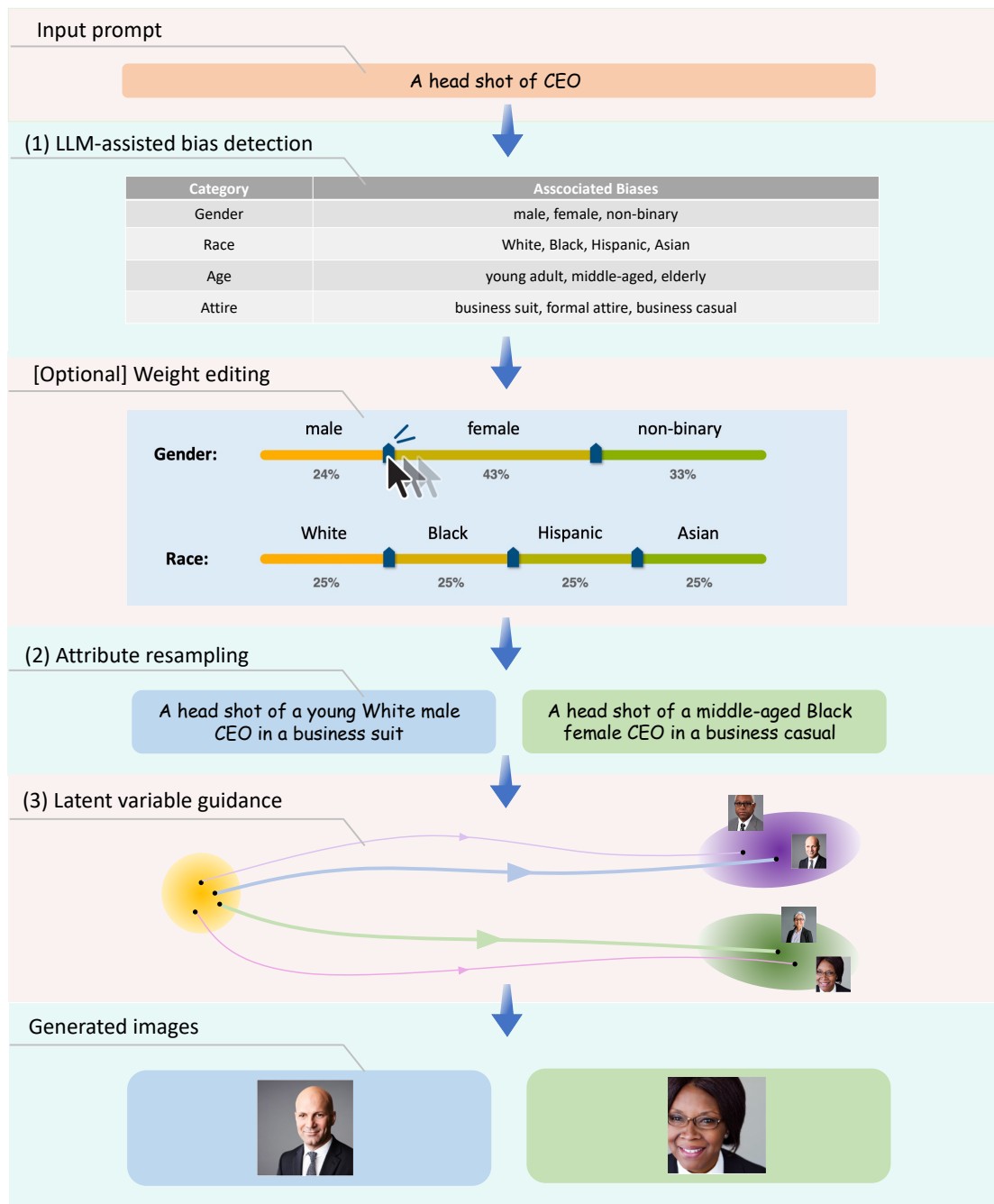

Figure 1: Overview of FairT2I. The LLM detects biases that exist in the input prompt and converts it into a set of bias-aware prompts by sampling the detected attributes from fair distributions. Users can interactively inspect, edit, and regenerate images, enabling both societal bias mitigation and improved diversity.

entirely at inference time without any model fine-tuning or pre-defined attributes, allowing users to directly intervene in the debiasing process. The framework consists of three main components:

1. LLM-based bias detection: Given a user prompt, the LLM analyzes the text to identify implicit categories and attributes that are likely to introduce bias in generated images, such as gender, ethnicity, or age. This transforms unstructured text into explicit bias dimensions that can be interpreted and controlled.

2. Attribute resampling: Once potential biases are identified, users can control the relative importance of each attribute (i.e., *attribute distribution*). The system supports multiple types of distributions, including a uniform distribution, one based on real-world statistics, and a custom user-specified distribution, enabling flexible control over fairness and diversity. Based on the selected distribution, the system resamples attributes to reflect the desired balance. This resampled distribution is then used to generate bias-aware prompts that explicitly encode the desired attribute composition.

3. Latent variable guidance: The image generation process is mathematically decomposed into attribute-conditioned components, and FairT2I reweights these components according to the adjusted attribute distribution. By leveraging the bias-aware prompts derived from attribute resampling, the framework realizes *latent variable guidance* in a way that is not ad hoc but theoretically grounded. This approach enables bias mitigation while maintaining fidelity and diversity in T2I, all without retraining the underlying diffusion model.

## 2 Related Work

**Text-to-image (T2I) models.** The generative capabilities of T2I models (Imagen-Team-Google et al., 2024; Esser et al., 2024; Chen et al., 2025a; Black Forest Labs, 2024) have improved dramatically through several key developments: training on large scale text image pairs (Schuhmann et al., 2022; Chen et al., 2023), architectural improvements (Peebles & Xie, 2023) from UNet (Ronneberger et al., 2015) to transformer-based designs (Vaswani et al., 2017; Dosovitskiy et al., 2020), and theoretical developments from diffusion models (Sohl-Dickstein et al., 2015; Ho et al., 2020; Song et al., 2020) to flow matching (Lipman et al., 2022; Liu et al., 2022; Lipman et al., 2024). Furthermore, recent fine tuning approaches have significantly reduced generation time by minimizing the required number of inference steps (Liu et al., 2023; Chen et al., 2024a; Sauer et al., 2024; 2023; Yang et al., 2023). A distinctive characteristic of state-of-the-art models is their use of separately trained components, such as Variational Auto Encoders (Kingma et al., 2014) for image compression, text encoders (Raffel et al., 2020; Radford et al., 2021), and flow model backbones, each trained on different datasets.

**Societal bias in T2I models.** Numerous studies have investigated biases in T2I models. Research such as Ghosh & Caliskan (2023); Wu et al. (2023); Bianchi et al. (2023) highlights the biases embedded in generated outputs for seemingly neutral input prompts that lack explicit identity- or demographic-related terms. Other works, including Cho et al. (2022); Luccioni et al. (2023) predefine sensitive human attributes and analyze biases in outputs generated from occupational input prompts. Naik & Nushi (2023) provides broader analyses, including comparative studies with statistical data or image search results, as well as spatial analyses of generated images. Wang et al. (2023) applies methods from social psychology to explore implicit and complex biases related to race and gender. Furthermore, Luccioni et al. (2023) introduces an interactive bias analysis tool leveraging clustering methods. Lastly, Chen et al. (2024b) examines the potential for AI-generated images to perpetuate harmful feedback loops, amplifying biases in AI systems when used as training data for future models. D'Incà et al. (2024); Chinchure et al. (2023) employ LLMs to detect open-ended biases in text-to-image models where users do not have to provide predefined bias attributes.

**Bias mitigation in T2I models.** One line of work focuses on making T2I models fair by fine-tuning or retraining various components, from the diffusion backbone (Kim et al., 2024) and text encoder Hirota et al. (2024) to postfix prompt embeddings Zhang et al. (2023) or additive residual image vectors Seth et al. (2023). While effective, such retraining is both time- and resource-intensive and must be repeated whenever societal notions of bias evolve. By contrast, several methods eliminate societal bias only by steering generation at inference time. FairDiffusion Friedrich et al. (2023) guides sampling with randomly chosen bias attributes; (Chuang et al., 2023) removes unwanted directions from the text embedding space via calibrated projection matrices; and ENTIGEN Bansal et al. (2022) simply appends an ethical fairness postfix to the input prompt. Although these approaches require no additional training, they do depend on a predefined list of bias categories and attribute values for control.

# 3 Methodologies

In this section, we present FairT2I, a mathematically grounded framework for mitigating societal biases in T2I generation. FairT2I introduces Latent Variable Guidance, which decomposes generation into attribute-conditioned components and enables explicit control over sensitive attributes. It further combines LLM-based bias detection to infer prompt-specific bias categories without predefined attribute sets, and attribute resampling to redefine the target attribute distribution, for example using real-world demographic statistics.

## 3.1 Preliminaries: Classifier-Free Guidance

In diffusion models (Ho et al., 2020; Sohl-Dickstein et al., 2015) and flow matching models (Esser et al., 2024; Lipman et al., 2022), classifier-free guidance (CFG) (Ho & Salimans, 2022) is a standard technique for sampling images $\mathbf{x}$ conditioned on a text prompt $\mathbf{y}$. CFG combines the unconditional score $\nabla_{\mathbf{x}} \log p(\mathbf{x})$ and the conditional score $\nabla_{\mathbf{x}} \log p(\mathbf{x} \mid \mathbf{y})$ to control the strength of text conditioning.

For a fixed text prompt $\mathbf{y}$, CFG can be interpreted as sampling from the tilted conditional distribution

$$p_\omega(\mathbf{x} \mid \mathbf{y}) \propto p(\mathbf{x})p(\mathbf{y} \mid \mathbf{x})^\omega,$$

where $\omega \in \mathbb{R}$ is the guidance scale, typically chosen as $\omega > 1$. By Bayes' rule, since $p(\mathbf{y})$ is constant with respect to $\mathbf{x}$, we have

$$p_\omega(\mathbf{x} \mid \mathbf{y}) \propto p(\mathbf{x})^{1-\omega}p(\mathbf{x} \mid \mathbf{y})^\omega.$$

Taking the score with respect to $\mathbf{x}$ gives

$$\nabla_{\mathbf{x}} \log p_\omega(\mathbf{x} \mid \mathbf{y}) = (1 - \omega)\nabla_{\mathbf{x}} \log p(\mathbf{x}) + \omega \nabla_{\mathbf{x}} \log p(\mathbf{x} \mid \mathbf{y}).$$

Equivalently, if we denote the unconditional and conditional scores by $s_{\mathrm{uncond}}(\mathbf{x}) = \nabla_{\mathbf{x}} \log p(\mathbf{x})$ and $s_{\mathrm{cond}}(\mathbf{x}, \mathbf{y}) = \nabla_{\mathbf{x}} \log p(\mathbf{x} \mid \mathbf{y})$, then the guided score is

$$s_{\mathrm{CFG}}(\mathbf{x}, \mathbf{y}) = s_{\mathrm{uncond}}(\mathbf{x}) + \omega\left(s_{\mathrm{cond}}(\mathbf{x}, \mathbf{y}) - s_{\mathrm{uncond}}(\mathbf{x})\right).$$

Thus, CFG can be viewed as a weighted combination of unconditional and text-conditional score components.

In the next subsection, we extend this score-guidance view by introducing a latent attribute variable $\mathbf{z}$. Instead of using a single text-conditional component, FairT2I combines attribute-conditioned components according to a target attribute distribution, which enables bias-aware control within the same guidance framework.

## 3.2 Latent Variable Guidance for Bias Mitigation

In latent variable guidance, societal bias is controlled by decomposing the network's predicted score function into a weighted sum of component score functions, each conditioned on a distinct latent variable. These weights are then adjusted to enforce fairness in the generated outputs. Central to our formulation is the following proposition.

**Proposition 1** (Posterior-weighted latent score decomposition)**.** *Let* $\mathbf{y}$ *denote the input text,* $\mathbf{x}$ *the generated image, and* $\mathbf{z}$ *a discrete latent variable taking values in a finite set* $\mathcal{Z}$*. The conditional score can be decomposed as*

$$\nabla_{\mathbf{x}} \log p(\mathbf{x} \mid \mathbf{y}) = \sum_{z \in \mathcal{Z}} p(\mathbf{z} = z \mid \mathbf{x}, \mathbf{y})\nabla_{\mathbf{x}} \log p(\mathbf{x} \mid \mathbf{z} = z, \mathbf{y}). \tag{1}$$

*Proof.* See Section A.1. □

Eq. (1) is an exact posterior-weighted decomposition. Directly using this identity during generation would require estimating $p(\mathbf{z} \mid \mathbf{x}_t, \mathbf{y})$ at each denoising step. FairT2I makes two modifications to obtain a practical bias-mitigation rule.

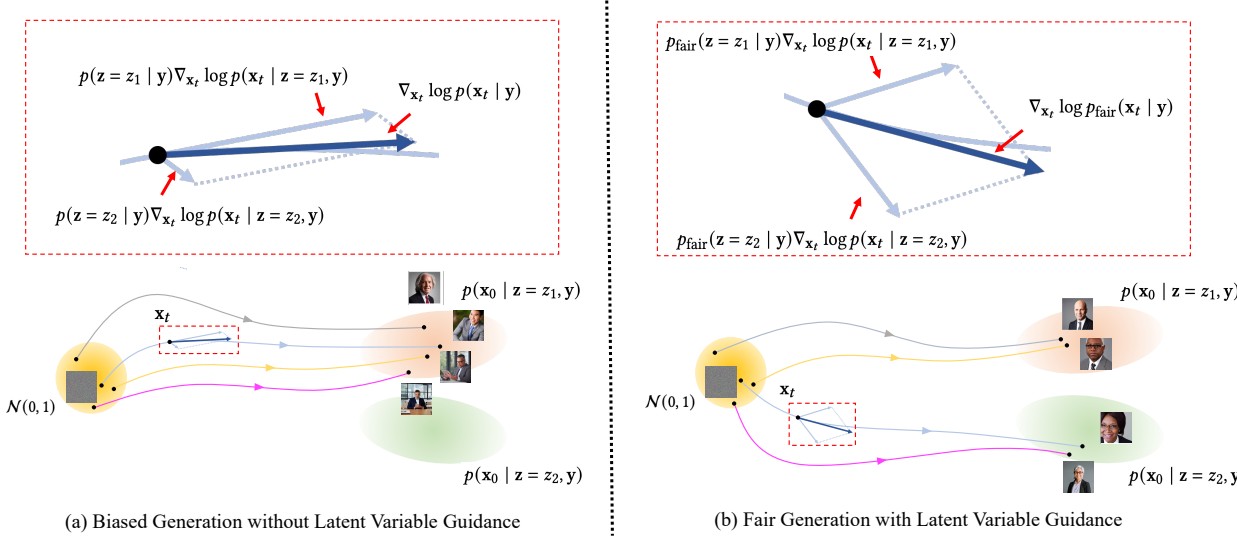

(a) Biased Generation without Latent Variable Guidance          (b) Fair Generation with Latent Variable Guidance

Figure 2: Comparison of generation processes without (left) and with (right) latent variable guidance for the input prompt $\mathbf{y} = A\ CEO$ and attribute set $\mathcal{Z} = \{z_1 = male, z_2 = female\}$. In the standard diffusion process, the model internalizes societal biases from the training data and denoises images toward male CEO representations, resulting in biased outputs. Latent variable guidance enables the adjustment of the denoising direction based on a fair attribute distribution, thereby allowing the mitigation of such biases in the generated results.

First, we use a high-noise approximation. When $\mathbf{x}_t$ is close to the Gaussian noise endpoint, it contains limited semantic information about the final image attributes. In this regime, the posterior attribute distribution is close to the text-conditioned prior, $p(\mathbf{z} \mid \mathbf{x}_t, \mathbf{y}) \approx p(\mathbf{z} \mid \mathbf{y})$. This approximation is justified formally in Section A.2. Although this approximation need not hold in later denoising stages, FairT2I applies the intervention through attribute-augmented text conditioning, $\mathbf{y} \oplus \mathbf{z}$, as described in Section 3.4. Since recent analyses suggest that text-based interventions have much weaker influence in later denoising stages (Liu et al., 2025), this limitation is expected to have a limited effect on the proposed guidance rule.

Second, we introduce a fairness intervention on the attribute distribution. Let $p_{\text{fair}}(\mathbf{z} \mid \mathbf{y})$ denote a target attribute distribution used for bias mitigation. This distribution replaces the model's implicit text-conditioned attribute prior $p(\mathbf{z} \mid \mathbf{y})$ in the guidance rule. This gives the prior-weighted FairT2I guidance score

$$s_{\text{FairT2I}}(\mathbf{x}_t, \mathbf{y}) := \sum_{z \in \mathcal{Z}} p_{\text{fair}}(\mathbf{z} = z \mid \mathbf{y}) \nabla_{\mathbf{x}_t} \log p_t(\mathbf{x}_t \mid \mathbf{z} = z, \mathbf{y}). \tag{2}$$

In this formulation, $p_{\text{fair}}(\mathbf{z} \mid \mathbf{y})$ serves as an intervention that replaces the model's implicit attribute prior in the text-conditioned guidance process, rather than estimating the original posterior attribute distribution throughout the denoising trajectory. The candidate attribute set $\mathcal{Z}$ is obtained by LLM-assisted bias detection, and the construction of $p_{\text{fair}}(\mathbf{z} \mid \mathbf{y})$ is described in Section 3.4.

This perspective allows us to contextualize existing bias mitigation strategies. Approaches such as replacing text encoders with fair alternatives (Hirota et al., 2024; Chuang et al., 2023), learning fair prompt embeddings (Zhang et al., 2023), or randomly sampling sensitive attributes (Friedrich et al., 2023) can all be interpreted as methods that implicitly or explicitly substitute the T2I model's original, potentially biased distribution $p_{\text{bias}}(\mathbf{z} \mid \mathbf{y})$ with a more equitable target distribution $p_{\text{fair}}(\mathbf{z} \mid \mathbf{y})$. Fig. 2 illustrates the generation process with latent variable guidance, compared to the standard process without it, for an input prompt $\mathbf{y} = A\ CEO$ and an attribute set $\mathcal{Z} = \{z_1 = male, z_2 = female\}$. In the standard setting, text-conditioned guidance is governed by the biased distribution $p_{\text{bias}}(\mathbf{z} \mid \mathbf{y})$, which tends to steer the generation toward male CEO representations. In contrast, latent variable guidance enables the replacement of $p_{\text{bias}}(\mathbf{z} \mid \mathbf{y})$ with a fair distribution $p_{\text{fair}}(\mathbf{z} \mid \mathbf{y})$, allowing the generation process to be steered toward more balanced and fair outcomes.

In practice, computing the summation over all possible values of $z$ in Eq. (2) can be computationally prohibitive, especially when dealing with a large or continuous space of latent attributes. To overcome this computational challenge, we employ Monte Carlo sampling from the fair distribution $p_{\text{fair}}(\mathbf{z} \mid \mathbf{y})$. Specifically, we approximate the expectation in Eq. (2) using a finite number of samples. For the simplest case, using an approach with a single sample, this approximation becomes:

$$s_{\text{FairT2I}}(\mathbf{x}_t, \mathbf{y}) \approx \nabla_{\mathbf{x}_t} \log p_t\big(\mathbf{x}_t \mid \widetilde{\mathbf{z}}, \mathbf{y}\big), \text{ where } \widetilde{\mathbf{z}} \sim p_{\text{fair}}(\mathbf{z} \mid \mathbf{y}).$$

### 3.3 LLM-Assisted Bias Detection

To implement latent variable guidance, we need to define a candidate set of biases $\mathcal{Z}$. The simplest approach is to predefine a closed set of biases, such as race and gender; however, this approach has several limitations. For instance, one significant limitation relates to computational challenges. The diversity of input prompts is virtually infinite, meaning that a predefined set of latent attributes $\mathcal{Z}$ can only appropriately handle a limited subset of these cases. Consequently, it is practically impossible to predefine a suitable $\mathcal{Z}$ for every conceivable prompt in advance. Another limitation concerns incomplete representation. If the latent attribute set $\mathcal{Z}$ is defined manually in a rule-based manner, it may fail to fully capture the diversity and context of the real world. This approach also risks overlooking biases embedded in the input text that are beyond human recognition.

To address these challenges, we leverage large language models (LLMs) (OpenAI et al., 2024; Anthropic, 2025) to automatically detect biases in the input text, as with existing bias detection methods (D'Incà et al., 2024; Chinchure et al., 2023). Specifically, we use the LLM to predict the set of possible latent attributes $\mathcal{Z}$ from the input text $\mathbf{y}$. LLMs are prompted to output a set of latent attributes that are likely to appear in images generated by T2I models with the input text in a JSON format. This approach allows us to handle a broader range of input prompts and to detect biases that may not be apparent to human annotators.

### 3.4 Attribute Resampling

When we have a set of latent attributes $\mathcal{Z}$ from LLM as potential biases in the input prompts, we can formulate $p_{\text{fair}}(\mathbf{z} = z \mid \mathbf{y})$ to perform sampling that mitigates bias. We consider two approaches to defining $p_{\text{fair}}(\mathbf{z} = z \mid \mathbf{y})$.

**Uniform distribution.** One simple approach is to set the distribution of latent attributes to be uniform across all possible values of $z$: $p_{\text{fair}}(\mathbf{z} = z \mid \mathbf{y}) = \frac{1}{|\mathcal{Z}|}$, which corresponds to simply mixing the scores $\nabla_{\mathbf{x}} \log p(\mathbf{x} \mid \mathbf{z} = z, \mathbf{y})$ with equal proportions across all possible values of $z$. This formulation coincides with that of Fair Diffusion (Friedrich et al., 2023).

**Employment statistics.** Research has shown that T2I models tend to exaggerate demographic stereotypes beyond what we observe in real-world distributions across various latent attributes (Naik & Nushi, 2023). One way to address this issue is to incorporate real-world statistical data as $p_{\text{fair}}(\mathbf{z} = z \mid \mathbf{y})$, ensuring that the generated image distributions are at least as balanced as real-world demographics.[1]

Upon sampling the latent attribute $z$ from the fair conditional distribution $p_{\text{fair}}(\mathbf{z} = z \mid \mathbf{y})$, we need to compute the score $\nabla_{\mathbf{x}} \log p(\mathbf{x} \mid \mathbf{z} = z, \mathbf{y})$. To enforce that the T2I model incorporates the sampled attribute $z$, we use an LLM to naturally fuse the textual input $\mathbf{y}$ and the latent attribute $\mathbf{z}$, and feed this augmented text $\mathbf{y} \oplus \mathbf{z}$ directly into the T2I model in place of $\mathbf{y}$.

## 4 Evaluation of LLM-Detected Bias Categories

This section evaluates whether LLMs can construct candidate bias tables suitable for FairT2I. We focus on two properties required by our method: broad coverage of possible bias categories and stable attribute construction across prompts.

---

[1]It is worth noting that real-world occupational distributions often reflect systemic biases and unequal access to opportunities, shaped by historical and societal factors such as limited access to education or workplace discrimination. These disparities highlight that real-world distributions are not inherently fair.

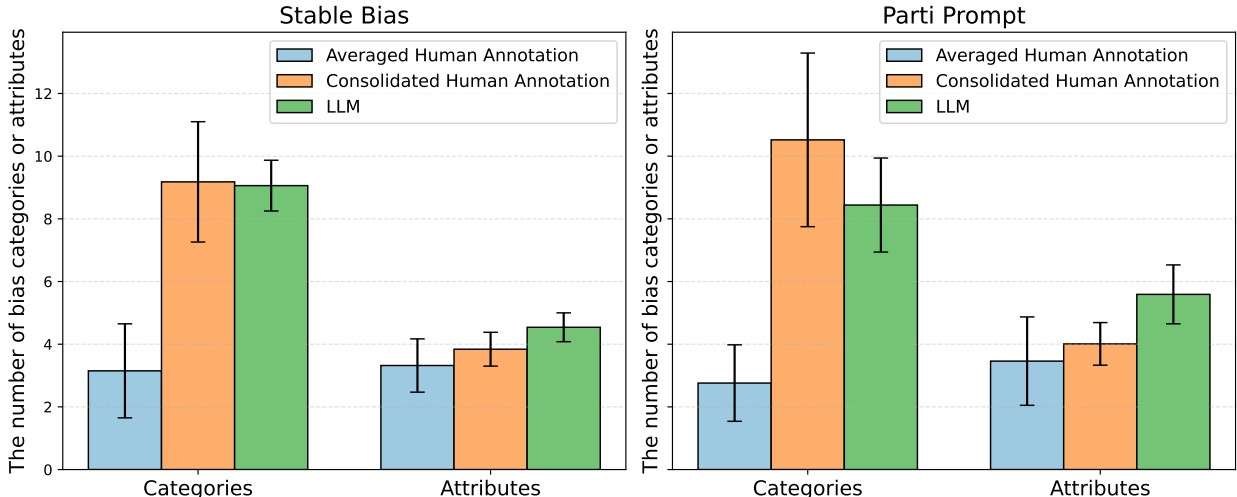

Figure 3: Number of detected categories and attributes per category for the Stable Bias (left) and Parti Prompt (right) datasets. Bars show Human Mean (blue), Human Union (orange), and LLM (green), with error bars indicating one standard deviation.

## 4.1 Experimental Settings

**Evaluation dataset.** We used two datasets for this study: one for assessing societal biases in professions and another for examining potential biases in more diverse textual prompts. To evaluate occupational bias, we used the Stable Bias profession dataset (Luccioni et al., 2023), which contains 131 occupations collected from the U.S. Bureau of Labor Statistics (BLS) (see Appendix A of Luccioni et al. (2023) for full details). To probe broader forms of bias, we adopt the Parti Prompt dataset (Yu et al., 2022), which includes more than 1,600 English prompts designed to evaluate text-to-image generation models and test their limitations. For efficiency, we randomly sample 50 prompts from each dataset, with the selected subset listed in Section B.1.

**LLMs for Bias Detection.** We used Claude 3.7-sonnet for LLM-assisted bias detection, following a small pilot study that compared GPT-4o OpenAI et al. (2024), Claude 3.7-sonnet Anthropic (2025), Llama 3.3 Grattafiori et al. (2024), and DeepSeek V3 Liu et al. (2024). Further details of this pilot study are provided in the appendix Section B.

## 4.2 Human Annotation and Reference Set Construction

To establish a human reference for bias categories, we conducted an annotation study using the same prompts as in the evaluation datasets. 100 text prompts, consisting of 50 prompts from the Stable Bias dataset (Luccioni et al., 2023) and 50 prompts from the PartiPrompts dataset (Yu et al., 2022) were divided into five groups of 20 prompts, with each group assigned to five independent annotators (25 participants in total). Annotators were instructed to list potential bias *categories* (e.g., gender, race, attire) along with multiple *attributes* per category (e.g., male, female, non-binary for gender). Clear format guidelines and illustrative examples were provided to ensure consistency. To maintain parity, the same few-shot examples were used both in the human instructions and in the system prompts provided to the LLMs.

For each prompt, we consolidated the five human responses by normalizing synonyms (e.g., *ethnicity → race*) and merging duplicates. The resulting union of all annotators produced a single consolidated human reference set of categories and attributes per prompt. Full details of the instructions, interface, and annotation guidelines are provided in Section B.3.

## 4.3 Results and Discussion

The goal of this evaluation is to assess the usefulness of LLM-generated candidate bias tables for downstream generation. Since FairT2I uses these tables to define latent attribute sets, the relevant question is whether the detected categories and attributes are sufficiently broad and consistent for controlled sampling.

Table 1: Gender and race attribute sets on the Stable Bias dataset. Few-shot examples show the categories provided in the annotation instructions and LLM prompt. Rows report the most frequent, minimum, and maximum sets for the human union and the LLM. Numbers in parentheses indicate the number of prompts in which the set occurred. For race, humans did not annotate the category in 4 prompts.

| | Gender | Race |
|---|---|---|
| **Few-shot examples** | non-binary, female, male | White, Asian, Black, Hispanic |
| **Human Union** | | |
| Most frequent | non-binary, female, male (29/50) | White, Asian, Black, Hispanic (16/50) |
| Minimum | non-binary, female, male | None |
| Maximum | non-binary, other, male, les, gay, female | Mixed, Asian, Hispanic, African-American, Arab, African, White, Native-American, Black |
| **LLM** | | |
| Most frequent | non-binary, female, male (46/50) | Middle Eastern, Asian, Hispanic, White, Indigenous, Black (13/50) |
| Minimum | non-binary, female, male | Middle Eastern, Asian, Hispanic, White, Black |
| Maximum | non-binary, female, androgynous, male | Middle Eastern, Asian, White, Hispanic/Latino, South Asian, Indigenous, Black |

**Categories and attributes.** To assess the ability to cover a broad range of bias categories and attributes, we compared the number of detected categories and attributes. Fig. 3 presents results for the Stable Bias dataset (left) and the Parti Prompt dataset (right), showing the average of five annotators (human mean; blue), the consolidated union of all annotators (human union; orange), and the LLM outputs (green). On the Stable Bias dataset, the human union and the LLM produced a comparable breadth of categories (9.18 vs. 9.06 on average), although the LLM generated more attributes per category (4.54 vs. 3.84). On the Parti Prompt dataset, however, humans (union) enumerated more categories than the LLM (10.52 vs. 8.44), while the LLM provided denser attribute lists (5.59 vs. 4.01). Notably, the human mean values are consistently lower than both the union and the LLM (for example, only 2.76 categories and 3.46 attributes in Parti Prompt), reflecting high inter-annotator variability. Taken together, these findings suggest that, while humans provide greater breadth when aggregated, LLMs offer more consistent depth within categories.

**Societal biases.** Table 1 illustrates the gender and race attribute sets detected on the Stable Bias dataset. On average, human consolidation yielded 3.52 gender attributes and 5.34 race attributes per prompt, while the LLM produced 3.08 and 5.8, respectively. Both humans and the LLM frequently reproduced the few-shot examples given in the annotation instructions, but diverged in their extensions: humans showed higher variability and occasionally omitted race altogether (4 prompts), whereas the LLM produced more systematic expansions that consistently included race.

Overall, the results show that LLM-based bias detection provides candidate attribute lists that are broader and more stable than those of an average individual annotator, while remaining comparable to the consolidated human union in the Stable Bias setting. This supports the use of LLMs as a scalable bias-table construction module in FairT2I.

## 5 Experiments: Societal Bias Mitigation

In this section, we evaluate FairT2I against existing methods using automatic metrics and human studies, demonstrating the detection of diverse textual biases and the corresponding mitigation in outputs.

### 5.1 Automatic Evaluation

**Evaluation dataset.** To evaluate societal bias in professions, we use the same Stable Bias profession dataset (Luccioni et al., 2023) introduced in Section 4, which contains 131 occupations from the U.S. Bureau of Labor Statistics (BLS). These two settings test whether FairT2I can steer generation toward different target attribute distributions. For the uniform distribution, we used the same subset of 50 randomly sampled occupations from this dataset as in Section 4. The selected subset of prompts is listed in Section B.1. For the BLS statistics, we use the five occupation prompts mentioned in Naik & Nushi (2023): *CEO*, *doctor*, *computer programmer*, *nurse*, and *house keeper*. As in Zhang et al. (2023), each profession is prefixed with "*A headshot of* " to form an input prompt.

**Evaluation metrics.** Following Hirota et al. (2024); Chuang et al. (2023), we quantify bias in generated outputs by computing statistical parity (Choi et al., 2020) between the empirical attribute distribution and an ideal target distribution. Consistent with Naik & Nushi (2023), we first detect faces in generated images using dlib (King, 2015), and then predict facial attributes using classifiers trained on the FairFace dataset (Karkkainen & Joo, 2021). Statistical parity is defined as the $\ell_2$ distance between two probability distributions. Formally, for two distributions $\mathbf{p} = (p_1, \ldots, p_n)$ and $\mathbf{q} = (q_1, \ldots, q_n)$ over $n$ outcomes, we define $\mathrm{SP}(\mathbf{p}, \mathbf{q}) = \|\mathbf{p} - \mathbf{q}\|_2 = \sqrt{\sum_{i=1}^{n}(p_i - q_i)^2}$. Lower statistical parity values indicate that the generated image distribution more closely matches the target fair distribution.

For each prompt in the evaluation subset, we generate 200 images, classify each image according to its FairFace attributes, and compute statistical parity. Additional implementation details are provided in Section C.1.

**Methods for comparison.** We compare FairT2I with existing debiasing approaches that do not require model finetuning. These include the unmodified Stable Diffusion model (Rombach et al., 2022a) without any bias mitigation (denoted as *Original*); ENTIGEN (Bansal et al., 2022), which appends an ethical injection phrase to the prompt; and Fair Diffusion (Friedrich et al., 2023), which randomly samples attributes from a predefined bias set and applies semantic guidance (Brack et al., 2023). Hyperparameter settings for ENTIGEN and Fair Diffusion are reported in Section C.2 and Section C.3, respectively. To ensure a fair comparison, all baseline and proposed methods use Stable Diffusion 1.5 (Rombach et al., 2022b) as the underlying text-to-image model.

**Implementation details.** We used Stable Diffusion 1.5 (Rombach et al., 2022b) as our T2I model. This choice was made because the hyperparameter settings and text encoders of the comparison methods were not compatible with the newer T2I architectures (Esser et al., 2024). We employed Claude 3.7-sonnet (Anthropic, 2025) for LLM-assisted bias detection, and GPT4o-mini (OpenAI et al., 2024) to fuse the textual input $\mathbf{y}$ with the sensitive attribute $\mathbf{z}$. The parameter settings for image generation and the exact prompts supplied to the LLM are detailed in Section C.4.

**Targeting uniform distribution.** Table 2 reports the SP scores comparing empirical and uniform distributions of gender and race in the Stable Bias Profession subset. FairT2I most closely matches the uniform baseline on both metrics. It demonstrates that, compared to existing methods, FairT2I generates data for each attribute with greater uniformity. See Section C for bar-chart comparisons and more details.

**Targeting BLS statistics.** Table 4 summarizes the gender and race SP scores between BLS statistics for *CEO*, *computer programmer*, *doctor*, *nurse*, and *housekeeper* prompts across four methods. FairT2I consistently achieves the lowest gender SP, outperforming ENTIGEN and FairDiffusion by large margins. For race SP, FairT2I yields the best scores on *CEO*, *computer programmer*, and *housekeeper*. These results confirm that FairT2I can also steer the generated distributions toward specified target distributions.

### 5.2 Monte Carlo Sample Ablation

We ablate the number of Monte Carlo samples $K$ used in Latent Variable Guidance, using the same automatic evaluation setup as in Section 5.1. We additionally report the per-image generation time measured on a single H200 GPU.

Table 2: Statistical Parity of Gender and Race Distributions by the original Stable Diffusion, ENTIGEN, FairDiffusion, and FairT2I (Ours).

| Method | Gender SP ↓ | Race SP ↓ |
|---|---|---|
| Original | 0.2056 | 0.3907 |
| ENTIGEN | 0.0661 | 0.2001 |
| FairDiffusion | 0.2083 | 0.3137 |
| FairT2I | **0.0123** | **0.1864** |

Table 3: Monte Carlo sample ablation for Latent Variable Guidance. Lower values are better.

| $K$ | sec/Img. | Gender SP↓ | Race SP↓ |
|---|---|---|---|
| 1 | **0.37** | 0.147 | **0.183** |
| 2 | 0.52 | **0.121** | 0.211 |
| 4 | 0.80 | 0.126 | 0.289 |
| 8 | 1.35 | 0.153 | 0.327 |

Table 4: Gender and race Statistical Parity (SP) scores between BLS statistics and generated images for CEO, computer programmer, doctor, nurse, and housekeeper. Lower is better. For each metric and occupation, the best score is shown in bold.

| Method | Gender SP ↓ | | | | | Race SP ↓ | | | | |
|---|---|---|---|---|---|---|---|---|---|---|
| | CEO | computer programmer | doctor | nurse | housekeeper | CEO | computer programmer | doctor | nurse | housekeeper |
| Original | 0.364 | 0.165 | 0.384 | 0.180 | 0.146 | 0.605 | 0.221 | 0.211 | 0.197 | 1.021 |
| ENTIGEN | 0.375 | 0.057 | 0.123 | 0.121 | **0.012** | 0.500 | 0.612 | 0.473 | 0.481 | 0.934 |
| FairDiffusion | 0.196 | 0.109 | 0.114 | 0.165 | 0.160 | 0.166 | 0.148 | **0.075** | **0.129** | 0.307 |
| FairT2I | **0.071** | **0.017** | **0.055** | **0.054** | **0.012** | **0.068** | **0.089** | 0.114 | 0.154 | **0.200** |

As shown in Table 3, $K = 1$ achieves the lowest Race SP and the fastest generation time, while $K = 2$ achieves the lowest Gender SP. Increasing $K$ does not consistently improve SP, despite reducing the variance of the Monte Carlo estimator for the target-weighted score average. This is because increasing $K$ averages multiple attribute-conditioned score directions within the same denoising trajectory. For semantically exclusive attributes such as gender or race, this can weaken the coherence of the attribute condition assigned to each image.

By contrast, $K = 1$ performs image-level attribute resampling: it samples one attribute from the target distribution $q(\mathbf{z} \mid \mathbf{y})$ and keeps it fixed throughout denoising. Thus, each image is generated under a coherent sampled attribute, while the aggregate distribution follows the target distribution in expectation.

## 6 Experiments: Diversity Control

In this section, we evaluate FairT2I not only for its ability to mitigate societal bias but also for its capacity to enhance the diversity of generated images. We conduct both quantitative (automatic) and qualitative (human) evaluations to assess the effectiveness of our method.

### 6.1 Experimental Settings

**Evaluation dataset.** We employ the Parti Prompt dataset (Yu et al., 2022), which comprises over 1,600 diverse English prompts intended to rigorously evaluate T2I generation models and probe their limitations. In this work, we focus on the uniform prompt distribution, as no particular target distribution is assumed. For practical efficiency, we randomly sample 50 prompts from the full dataset as in Section 4. The selected subset of prompts is listed in Section B.1.

**Models and baselines.** We compare FairT2I against classifier-free guidance (CFG) (Ho & Salimans, 2022) at three guidance scales (7.0, 4.0, 1.0), which modulate the trade-off between image fidelity and diversity.

**Implementation details.** We use Stable Diffusion 3.5-large (Esser et al., 2024) as the T2I model. We employ Claude 3.7-sonnet (Anthropic, 2025) for LLM-assisted bias detection, and GPT-4o-mini (OpenAI et al., 2024) to fuse textual input $\mathbf{y}$ with sensitive attribute $\mathbf{z}$. Generation hyperparameters and LLM prompts are in Section D.2.

Table 5: Comparison of classifier-free guidance (CFG) at guidance scales 7.0, 4.0, and 1.0 and of FairT2I on the FID, CLIPScore, CLIP Trace, and BLIP2 Trace metrics. A guidance scale of 1.0 corresponds to generation without CFG.

| Method | FID ↓ | CLIPScore ↑ | CLIP Trace ↑ | BLIP2 Trace ↑ |
|---|---|---|---|---|
| CFG-7.0 | 27.13 | 29.84 | 11.66 | 247.01 |
| CFG-4.0 | 26.46 | 30.09 | 11.87 | 250.24 |
| No CFG (CFG-1.0) | 34.47 | 28.06 | 21.05 | 473.64 |
| FairT2I (Ours) | 26.24 | 28.35 | 19.18 | 454.49 |

## 6.2 Automatic Evaluation

**Automatic metrics and generation protocol.** To assess image fidelity, we compute the Fréchet Inception Distance (FID) (Heusel et al., 2017) on the COCO Karpathy test split (Lin et al., 2014). To evaluate text–image alignment, we report CLIPScore (Hessel et al., 2021) between each generated image and its input prompt. To quantify diversity, we embed all generated images into CLIP (Radford et al., 2021) and BLIP2 (Li et al., 2023) feature spaces, compute the covariance of embeddings, and use its trace as the diversity metric. For each prompt, we generate 200 images and compute these metrics accordingly.

**Results.** Table 5 presents quantitative evaluations in terms of FID, CLIPScore, CLIP Trace and BLIP2 Trace, illustrating the trade-off between image fidelity, text alignment and diversity. Under CFG 7.0 and CFG 4.0, FID values are low but Trace metrics remain small. By contrast, CFG 1.0 yields a high Trace value at the expense of elevated FID. Unlike these existing methods, FairT2I achieves low FID and high Trace without substantially reducing CLIPScore. While classifier-free guidance cannot achieve both image quality and diversity, FairT2I succeeds on both measures.

## 6.3 Human Evaluation

To complement the automatic metrics, we conducted a human evaluation to assess the perceptual diversity, image quality, and text–image alignment of FairT2I in comparison with baseline methods.

**Setup.** For each prompt in the 50-prompt subset of the Parti Prompt dataset, we generated nine images using each of four methods: CFG-1.0, CFG-4.0, CFG-7.0, and FairT2I. All images were arranged in $3 \times 3$ grids per method, enabling side-by-side comparison across methods.

**Task design.** To keep the annotation time manageable, the 50 prompts were divided into five tasks, each containing ten prompts and the corresponding generations from all four methods. Each task required roughly 20 minutes to complete. We recruited a total of 100 annotators, resulting in 20 independent raters per prompt.

**Procedure.** Annotators were asked to rate every image on a five-point absolute Likert scale (1 = worst, 5 = best) along three criteria: **Diversity**, which measures how different the multiple images from the same model look for the same prompt, with high diversity indicating clearly different variations in viewpoint, layout, or details and low diversity indicating nearly identical images; **Image Quality**, which measures how clear, coherent, and visually convincing the image looks within its intended style, with high-quality images being sharp, well-structured, and free from obvious AI artifacts such as severe distortions, broken anatomy, or unnatural textures, even when the content is fantastical or stylized; and **Text Alignment**, which measures how well the generated image matches the prompt in terms of objects, attributes, and relationships.

Detailed screenshots of the annotation interface, instructions, and answer forms are provided in Appendix D.5.

**Results.** Table 6 and Table 7 summarize the outcomes of the human evaluation. Inter-annotator agreement was modest but acceptable for aggregation (Krippendorff's $\alpha$: Diversity 0.241, Quality 0.141, Alignment 0.136). FairT2I achieved the highest mean ratings in both *diversity* and *quality* among all compared methods.

Table 6: Human evaluation summary statistics for each task and model.

| Criteria | Model | Mean ↑ | Std. |
|---|---|---|---|
| Diversity | CFG-1.0 | 3.416 | 1.120 |
| | CFG-4.0 | 2.697 | 1.154 |
| | CFG-7.0 | 2.735 | 1.226 |
| | FairT2I | **3.643** | 1.105 |
| Quality | CFG-1.0 | 2.775 | 1.152 |
| | CFG-4.0 | 3.686 | 0.934 |
| | CFG-7.0 | 3.751 | 1.022 |
| | FairT2I | **3.827** | 0.991 |
| Alignment | CFG-1.0 | 3.152 | 1.214 |
| | CFG-4.0 | **3.833** | 1.051 |
| | CFG-7.0 | 3.862 | 1.096 |
| | FairT2I | 3.682 | 1.123 |

Table 7: One-sided Mann–Whitney U test results ($H_1$: FairT2I > Baseline). Findings indicate whether FairT2I achieved a significantly higher score ($p < 0.001$).

| Criteria | Baseline | p-value |
|---|---|---|
| Diversity | CFG-1.0 | $3.54 \times 10^{-6}$ |
| | CFG-4.0 | $5.90 \times 10^{-68}$ |
| | CFG-7.0 | $6.66 \times 10^{-60}$ |
| Quality | CFG-1.0 | $1.44 \times 10^{-87}$ |
| | CFG-4.0 | $4.74 \times 10^{-5}$ |
| | CFG-7.0 | $3.93 \times 10^{-2}$ |
| Alignment | CFG-1.0 | $4.97 \times 10^{-23}$ |
| | CFG-4.0 | $9.98 \times 10^{-1}$ |
| | CFG-7.0 | $1.00 \times 10^{0}$ |

For diversity, FairT2I obtained a mean score of $3.64 \pm 1.10$, which was substantially higher than CFG-1.0 (3.42), CFG-4.0 (2.70), and CFG-7.0 (2.74). All corresponding one-sided Mann–Whitney U tests indicated statistically significant improvements ($p < 0.001$), confirming that FairT2I produced perceptibly more diverse outputs than any baseline. For image quality, FairT2I also slightly outperformed all CFG variants ($3.83\pm0.99$ vs. 2.78–3.75); the improvements over CFG-1.0 and CFG-4.0 were significant ($p < 0.001$ and $p = 4.7 \times 10^{-5}$, respectively), while the difference from CFG-7.0 was not significant ($p = 0.039$). In text–image alignment, FairT2I maintained competitive performance ($3.68 \pm 1.12$), outperforming CFG-1.0 significantly ($p < 0.001$) but not CFG-4.0 or CFG-7.0 ($p > 0.9$). Overall, these results demonstrate that FairT2I improves perceived diversity and quality without compromising alignment, achieving a favorable balance between fidelity and variability that is consistent with the automatic evaluation trends.

## 7 Limitations

**LLM bottlenecks.** As shown in Fig. 7, FairT2I generates fewer images for *Southeast Asian* and *Indian* categories compared to other classes. This is partly because the LLM does not detect *Southeast Asian* or *Indian* as race categories. Notably, in the user study reported in Section 4, only 3 out of 25 participants included *Indian* in the *race* category, and none mentioned *Southeast Asian*. This suggests that such categories are underrepresented not only in LLM outputs but also in human conceptualization, which can further reinforce detection biases. By explicitly supplying these attributes as few-shot examples in the LLM prompt, this issue can be mitigated.

**Text encoder bottlenecks.** FairT2I reflects latent attributes in image generation via explicit prompt modifications. As demonstrated by Hirota et al. (2024), information such as gender that is explicitly specified is accurately captured by the text encoder in the generated images. However, long prompts and complex logical relations (Tang et al., 2023; Zhang et al., 2023) may not be reflected in the outputs. This limitation could potentially be addressed by using a long-context model (Zhang et al., 2024a) or a vision-language model-based text encoder (Chen et al., 2025b).

## 8 Conclusions

In this paper, we have presented FairT2I, a novel approach to debiasing T2I models by decomposing score functions using latent variable guidance and leveraging LLMs for bias detection and attribute resampling. We confirmed FairT2I can effectively debias T2I models without the need for model fine-tuning or tedious hyperparameter tuning. Furthermore, FairT2I can be used not only for societal bias mitigation but also to improve and control diversity.

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

# A   Theoretical Justification of Latent Variable Guidance

## A.1   Posterior-Weighted Score Decomposition

**Proposition 1.** *Let $\mathbf{y}$ denote the input text, $\mathbf{x}_t$ the noisy image latent at timestep $t$, and $\mathbf{z}$ a discrete latent variable taking values in a finite set $\mathcal{Z}$. The conditional score admits the following posterior-weighted decomposition:*

$$\nabla_{\mathbf{x}_t} \log p_t(\mathbf{x}_t \mid \mathbf{y}) = \sum_{z \in \mathcal{Z}} p(\mathbf{z} = z \mid \mathbf{x}_t, \mathbf{y}) \nabla_{\mathbf{x}_t} \log p_t(\mathbf{x}_t \mid \mathbf{z} = z, \mathbf{y}). \tag{3}$$

*Proof.* We prove the identity on the support where the relevant densities are positive. Since $\mathbf{z}$ takes values in a finite set $\mathcal{Z}$, we can marginalize over $\mathbf{z}$ as

$$p_t(\mathbf{x}_t \mid \mathbf{y}) = \sum_{z \in \mathcal{Z}} p_t(\mathbf{x}_t, \mathbf{z} = z \mid \mathbf{y}) \tag{4}$$

$$= \sum_{z \in \mathcal{Z}} p_t(\mathbf{x}_t \mid \mathbf{z} = z, \mathbf{y}) p(\mathbf{z} = z \mid \mathbf{y}). \tag{5}$$

Taking the score of this marginal distribution gives

$$\nabla_{\mathbf{x}_t} \log p_t(\mathbf{x}_t \mid \mathbf{y}) = \frac{\nabla_{\mathbf{x}_t} p_t(\mathbf{x}_t \mid \mathbf{y})}{p_t(\mathbf{x}_t \mid \mathbf{y})} \tag{6}$$

$$= \frac{\sum_{z \in \mathcal{Z}} p(\mathbf{z} = z \mid \mathbf{y}) \nabla_{\mathbf{x}_t} p_t(\mathbf{x}_t \mid \mathbf{z} = z, \mathbf{y})}{p_t(\mathbf{x}_t \mid \mathbf{y})}. \tag{7}$$

By Bayes' rule,

$$p(\mathbf{z} = z \mid \mathbf{x}_t, \mathbf{y}) = \frac{p(\mathbf{z} = z \mid \mathbf{y}) p_t(\mathbf{x}_t \mid \mathbf{z} = z, \mathbf{y})}{p_t(\mathbf{x}_t \mid \mathbf{y})}. \tag{8}$$

Equivalently,

$$\frac{p(\mathbf{z} = z \mid \mathbf{y})}{p_t(\mathbf{x}_t \mid \mathbf{y})} = \frac{p(\mathbf{z} = z \mid \mathbf{x}_t, \mathbf{y})}{p_t(\mathbf{x}_t \mid \mathbf{z} = z, \mathbf{y})}. \tag{9}$$

Substituting this into the score expression yields

$$\nabla_{\mathbf{x}_t} \log p_t(\mathbf{x}_t \mid \mathbf{y}) = \sum_{z \in \mathcal{Z}} p(\mathbf{z} = z \mid \mathbf{x}_t, \mathbf{y}) \frac{\nabla_{\mathbf{x}_t} p_t(\mathbf{x}_t \mid \mathbf{z} = z, \mathbf{y})}{p_t(\mathbf{x}_t \mid \mathbf{z} = z, \mathbf{y})} \tag{10}$$

$$= \sum_{z \in \mathcal{Z}} p(\mathbf{z} = z \mid \mathbf{x}_t, \mathbf{y}) \nabla_{\mathbf{x}_t} \log p_t(\mathbf{x}_t \mid \mathbf{z} = z, \mathbf{y}). \tag{11}$$

This proves the posterior-weighted score decomposition. □

## A.2   High-Noise Prior Approximation

**Lemma 1** (High-noise prior approximation). *Fix a prompt $\mathbf{y}$. Suppose that the forward process is*

$$\mathbf{X}_t = a_t \mathbf{X}_0 + \sigma_t \boldsymbol{\epsilon}, \qquad \boldsymbol{\epsilon} \sim \mathcal{N}(0, I), \tag{12}$$

*where $\sigma_t > 0$, $\boldsymbol{\epsilon}$ is independent of $\mathbf{X}_0$, and $\mathbf{X}_0$ has finite conditional covariance*

$$\Sigma_{\mathbf{y}} = \mathrm{Cov}(\mathbf{X}_0 \mid \mathbf{Y} = \mathbf{y}). \tag{13}$$

*Assume further that $\mathbf{Z} \to \mathbf{X}_0 \to \mathbf{X}_t$ forms a Markov chain conditioned on $\mathbf{Y} = \mathbf{y}$. Then*

$$\mathbb{E}_{\mathbf{x}_t \sim p_t(\cdot \mid \mathbf{y})} \left[ \mathrm{TV} \left( p(\mathbf{z} \mid \mathbf{x}_t, \mathbf{y}), p(\mathbf{z} \mid \mathbf{y}) \right) \right] \leq \frac{|a_t|}{2\sigma_t} \sqrt{\mathrm{tr}(\Sigma_{\mathbf{y}})}. \tag{14}$$

*Thus, as the signal-to-noise ratio $a_t^2 / \sigma_t^2$ approaches zero, the posterior attribute distribution $p(\mathbf{z} \mid \mathbf{x}_t, \mathbf{y})$ approaches the text-conditioned prior $p(\mathbf{z} \mid \mathbf{y})$ on average.*

*Proof.* We fix the prompt $\mathbf{y}$ throughout the proof. Since $\mathbf{Z} \to \mathbf{X}_0 \to \mathbf{X}_t$ forms a Markov chain conditioned on $\mathbf{Y} = \mathbf{y}$, the data processing inequality gives

$$I(\mathbf{Z}; \mathbf{X}_t \mid \mathbf{Y} = \mathbf{y}) \leq I(\mathbf{X}_0; \mathbf{X}_t \mid \mathbf{Y} = \mathbf{y}). \tag{15}$$

We now upper bound the right-hand side. The forward process

$$\mathbf{X}_t = a_t \mathbf{X}_0 + \sigma_t \boldsymbol{\epsilon} \tag{16}$$

is an additive Gaussian channel with noise covariance $\sigma_t^2 I$. Using the maximum-entropy property of the Gaussian distribution under a fixed covariance constraint (Cover & Thomas, 2006), we have

$$I(\mathbf{X}_0; \mathbf{X}_t \mid \mathbf{Y} = \mathbf{y}) = h(\mathbf{X}_t \mid \mathbf{Y} = \mathbf{y}) - h(\mathbf{X}_t \mid \mathbf{X}_0, \mathbf{Y} = \mathbf{y}) \tag{17}$$

$$\leq \frac{1}{2} \log \left( (2\pi e)^d \det(a_t^2 \Sigma_\mathbf{y} + \sigma_t^2 I) \right) - \frac{1}{2} \log \left( (2\pi e)^d \det(\sigma_t^2 I) \right) \tag{18}$$

$$= \frac{1}{2} \log \det \left( I + \frac{a_t^2}{\sigma_t^2} \Sigma_\mathbf{y} \right). \tag{19}$$

Here, $d$ is the dimension of $\mathbf{X}_0$. The inequality follows because

$$\mathrm{Cov}(\mathbf{X}_t \mid \mathbf{Y} = \mathbf{y}) = a_t^2 \Sigma_\mathbf{y} + \sigma_t^2 I,$$

and a Gaussian distribution has the largest differential entropy among all continuous distributions with the same covariance.

Since $\Sigma_\mathbf{y}$ is positive semidefinite, $\frac{a_t^2}{\sigma_t^2} \Sigma_\mathbf{y}$ is also positive semidefinite. Using $\log \det(I + A) \leq \mathrm{tr}(A)$ for positive semidefinite $A$, we obtain

$$I(\mathbf{X}_0; \mathbf{X}_t \mid \mathbf{Y} = \mathbf{y}) \leq \frac{a_t^2}{2\sigma_t^2} \mathrm{tr}(\Sigma_\mathbf{y}). \tag{20}$$

Combining this bound with Eq. (15) gives

$$I(\mathbf{Z}; \mathbf{X}_t \mid \mathbf{Y} = \mathbf{y}) \leq \frac{a_t^2}{2\sigma_t^2} \mathrm{tr}(\Sigma_\mathbf{y}). \tag{21}$$

Next, we rewrite the conditional mutual information as an expected KL divergence:

$$I(\mathbf{Z}; \mathbf{X}_t \mid \mathbf{Y} = \mathbf{y}) = \mathbb{E}_{\mathbf{z}, \mathbf{x}_t \sim p(\cdot, \cdot \mid \mathbf{y})} \left[ \log \frac{p(\mathbf{z}, \mathbf{x}_t \mid \mathbf{y})}{p(\mathbf{z} \mid \mathbf{y}) p(\mathbf{x}_t \mid \mathbf{y})} \right] \tag{22}$$

$$= \mathbb{E}_{\mathbf{z}, \mathbf{x}_t \sim p(\cdot, \cdot \mid \mathbf{y})} \left[ \log \frac{p(\mathbf{z} \mid \mathbf{x}_t, \mathbf{y})}{p(\mathbf{z} \mid \mathbf{y})} \right] \tag{23}$$

$$= \mathbb{E}_{\mathbf{x}_t \sim p_t(\cdot \mid \mathbf{y})} \left[ \mathrm{KL} \left( p(\mathbf{z} \mid \mathbf{x}_t, \mathbf{y}) \,\|\, p(\mathbf{z} \mid \mathbf{y}) \right) \right]. \tag{24}$$

By Pinsker's inequality,

$$\mathrm{TV} \left( p(\mathbf{z} \mid \mathbf{x}_t, \mathbf{y}), p(\mathbf{z} \mid \mathbf{y}) \right) \leq \sqrt{\frac{1}{2} \mathrm{KL} \left( p(\mathbf{z} \mid \mathbf{x}_t, \mathbf{y}) \,\|\, p(\mathbf{z} \mid \mathbf{y}) \right)}. \tag{25}$$

Taking the expectation over $\mathbf{x}_t \sim p_t(\cdot \mid \mathbf{y})$ gives

$$\mathbb{E}_{\mathbf{x}_t \sim p_t(\cdot \mid \mathbf{y})} \left[ \mathrm{TV} \left( p(\mathbf{z} \mid \mathbf{x}_t, \mathbf{y}), p(\mathbf{z} \mid \mathbf{y}) \right) \right] \tag{26}$$

$$\leq \mathbb{E}_{\mathbf{x}_t \sim p_t(\cdot \mid \mathbf{y})} \left[ \sqrt{\frac{1}{2} \mathrm{KL} \left( p(\mathbf{z} \mid \mathbf{x}_t, \mathbf{y}) \,\|\, p(\mathbf{z} \mid \mathbf{y}) \right)} \right]. \tag{27}$$

Since the square-root function is concave, Jensen's inequality implies

$$\mathbb{E}_{\mathbf{x}_t \sim p_t(\cdot|\mathbf{y})} \left[ \sqrt{\frac{1}{2} \mathrm{KL}\left(p(\mathbf{z} \mid \mathbf{x}_t, \mathbf{y}) \,\|\, p(\mathbf{z} \mid \mathbf{y})\right)} \right] \tag{28}$$

$$\leq \sqrt{\frac{1}{2} \mathbb{E}_{\mathbf{x}_t \sim p_t(\cdot|\mathbf{y})} \left[ \mathrm{KL}\left(p(\mathbf{z} \mid \mathbf{x}_t, \mathbf{y}) \,\|\, p(\mathbf{z} \mid \mathbf{y})\right) \right]} \tag{29}$$

$$= \sqrt{\frac{1}{2} I(\mathbf{Z}; \mathbf{X}_t \mid \mathbf{Y} = \mathbf{y})}. \tag{30}$$

Using Eq. (21), we finally obtain

$$\mathbb{E}_{\mathbf{x}_t \sim p_t(\cdot|\mathbf{y})} \left[ \mathrm{TV}\left(p(\mathbf{z} \mid \mathbf{x}_t, \mathbf{y}), p(\mathbf{z} \mid \mathbf{y})\right) \right] \leq \sqrt{\frac{1}{2} \cdot \frac{a_t^2}{2\sigma_t^2} \mathrm{tr}(\Sigma_{\mathbf{y}})} \tag{31}$$

$$= \frac{|a_t|}{2\sigma_t} \sqrt{\mathrm{tr}(\Sigma_{\mathbf{y}})}. \tag{32}$$

This proves the claim. $\qquad\square$

## B  Details for Bias Detection Experiment

### B.1  Evaluation Dataset Details

For efficient evaluation, we used a subset of 50 randomly selected prompts from each dataset: profession names from the Stable Bias dataset (Luccioni et al., 2023) and textual prompts from the Parti Prompt dataset (Yu et al., 2022). The selected subsets are listed in Table 8 and Table 10, respectively.

Table 8: A 50 profession subset of the stable bias professions dataset used in the experiment. The subset is randomly selected from the full dataset.

| Professions | | | |
|---|---|---|---|
| aerospace engineer | aide | author | bartender |
| carpenter | cashier | CEO | civil engineer |
| cleaner | coach | compliance officer | cook |
| dental assistant | dentist | detective | electrical engineer |
| engineer | facilities manager | fast food worker | file clerk |
| graphic designer | hairdresser | head cook | health technician |
| industrial engineer | interior designer | interviewer | inventory clerk |
| jailer | machinery mechanic | manicurist | massage therapist |
| medical records specialist | mental health counselor | metal worker | office clerk |
| painter | payroll clerk | physical therapist | plane mechanic |
| postal worker | psychologist | purchasing agent | repair worker |
| roofer | sales manager | sheet metal worker | social worker |
| underwriter | welder | | |

Table 9: Gender and race distributions from BLS 2024 statistics. Race percentages have been normalized so that they sum to 100 %.

| Occupation | Female (%) | Male (%) | White (%) | Black (%) | Asian (%) | Hispanic (%) |
|---|---|---|---|---|---|---|
| CEO | 33.0 | 67.0 | 82.2 | 5.8 | 6.1 | 5.8 |
| Doctor | 44.5 | 55.5 | 64.6 | 7.0 | 22.2 | 6.2 |
| Computer Programmer | 17.8 | 82.2 | 65.7 | 8.3 | 16.0 | 10.1 |
| Nurse | 86.8 | 13.2 | 67.0 | 14.7 | 9.1 | 9.1 |
| Housekeeper | 87.7 | 12.3 | 51.3 | 10.2 | 3.1 | 35.3 |

Table 10: A 50 prompt subset of the Parti Prompt dataset used in the experiment. The subset is randomly selected from the full dataset.

| No. | Description | No. | Description |
|---|---|---|---|
| 1. | a pile of cash on a wooden table | 26. | a paranoid android freaking out and jumping into the air because it is surrounded by colorful Easter eggs |
| 2. | five frosted glass bottles | 27. | a musical note |
| 3. | a view of the Big Dipper in the night sky | 28. | Anubis wearing sunglasses and sitting astride a hog motorcycle |
| 4. | a cow | 29. | a photograph of a squirrel holding an arrow above its head and holding a longbow in its left hand |
| 5. | a Ferrari Testarossa in front of the Kremlin | 30. | an elder politician giving a campaign speech |
| 6. | a rowboat | 31. | a hot air balloon with a yin-yang symbol |
| 7. | a portrait of a postal worker who has forgotten their mailbag | 32. | overhead view of three people looking down at the street from the top of a tall building |
| 8. | a helicopter hovering over Times Square | 33. | meaning of life |
| 9. | a t-shirt with Carpe Diem written on it | 34. | a laptop no letters on its keyboard |
| 10. | a canal in Venice | 35. | a glass of orange juice with an orange peel stuck on the rim |
| 11. | trees seen through a car window on a rainy day | 36. | a yellow tiger with blue stripes |
| 12. | a grumpy porcupine handing a check for $10 | 37. | cash |
| 13. | a watermelon chair | 38. | matching socks with cute cats on them |
| 14. | a triangle with a smiling face | 39. | five chairs |
| 15. | Four deer surrounding a moose. | 40. | an airplane flying into a cloud that looks like monster |
| 16. | an old red truck parked by the geyser Old Faithful | 41. | a long-island ice tea cocktail next to a napkin |
| 17. | a tree surrounded by flowers | 42. | a white robot passing a soccer ball to a red robot |
| 18. | A photo of a Ming Dynasty vase on a leather topped table. | 43. | a black towel |
| 19. | a chair | 44. | a capybara |
| 20. | a wooden post | 45. | artificial intelligence |
| 21. | Jupiter rises on the horizon. | 46. | Portrait of a tiger wearing a train conductor's hat and holding a skateboard that has a yin-yang symbol on it. charcoal sketch |
| 22. | A raccoon wearing formal clothes | 47. | Anime illustration of the Great Pyramid sitting next to the Parthenon under a blue night sky of roiling energy |
| 23. | A bowl of soup that looks like a monster with tofu says deep learning | 48. | a chess queen to the right of a chess knight |
| 24. | the Mona Lisa in the style of Minecraft | 49. | a Tyrannosaurus Rex roaring in front of a palm tree |
| 25. | a sphere | 50. | Portrait of a gecko wearing a train conductor's hat and holding a flag that has a yin-yang symbol on it. Child's crayon drawing. |

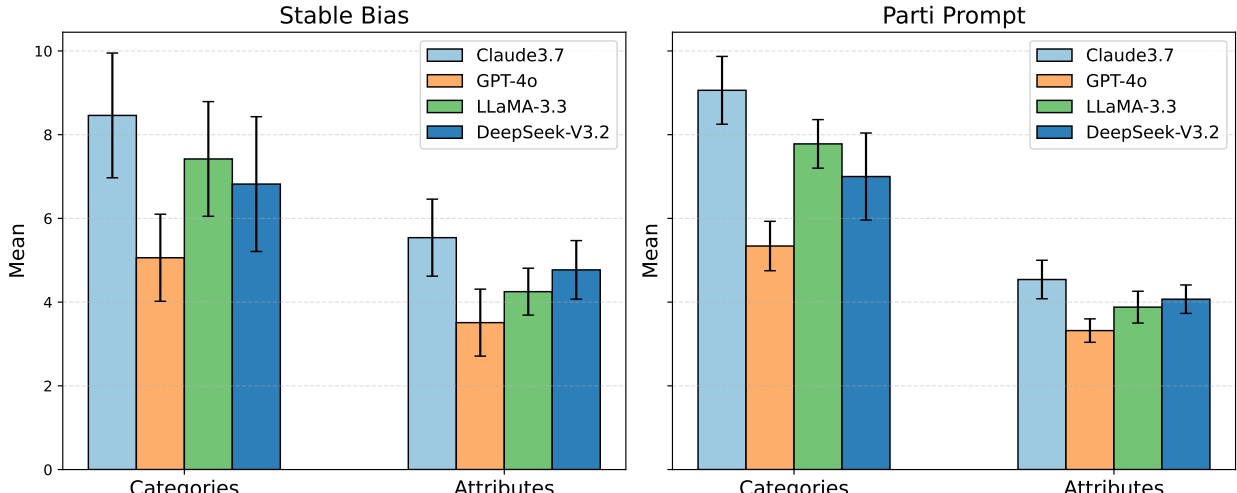

Figure 4: The number of detected categories and attributes per category for Stable Bias (left) and Parti Prompt (right) datasets. Bars show Claude3.7-sonnet (light blue), GPT-4o (orange), LlaMA-3.3 (green), and DeepSeek-V3.2 (blue) with error bars indicating one standard deviation.

## B.2 Pilot Study Details

We conducted a pilot comparison of four LLMs: GPT-4o OpenAI et al. (2024), Claude-3.7-Sonnet Anthropic (2025), LLaMA-3.3 Grattafiori et al. (2024), and DeepSeek-V3 Liu et al. (2024), to assess their ability to detect potential bias-relevant categories and attributes in text prompts. The comparison was performed using two evaluation subsets: 50 prompts sampled from the Stable Bias dataset and another 50 prompts from the Parti Prompt dataset, as listed in Section B.1.

For each model, we calculated the average number of detected categories per prompt and the average number of attributes per category. As summarized in Fig. 4, Claude3.7-Sonnet consistently detected a broader range of bias-relevant categories and attributes compared to the other LLMs, followed by LLaMA-3.3 and DeepSeek-V3, while GPT-4o showed the most conservative detection behavior.

Quantitatively, on the Stable Bias subset, Claude detected $8.46 \pm 1.49$ categories with $5.54 \pm 0.92$ attributes per category, outperforming GPT-4o ($5.06 \pm 1.04$ and $3.51 \pm 0.80$, respectively), LLaMA 3.3 ($7.42 \pm 1.37$ and $4.25 \pm 0.56$), and DeepSeek-V3 ($6.82 \pm 1.61$ and $4.77 \pm 0.70$). Similarly, on the Parti Prompt subset, Claude achieved $9.06 \pm 0.81$ categories and $4.54 \pm 0.46$ attributes per category, again exceeding GPT-4o ($5.34 \pm 0.59$, $3.32 \pm 0.28$), LLaMA-3.3 ($7.78 \pm 0.58$, $3.88 \pm 0.38$), and DeepSeek-V3 ($7.00 \pm 1.04$, $4.07 \pm 0.34$).

Table 11 further illustrates qualitative differences among the models using the prompt "*A CEO*" as an example. Claude 3.7 Sonnet identifies a richer set of categories, such as *body type*, *accessories*, and *facial expression*, which were often omitted by other models. This broader detection suggests that Claude may capture subtler implicit biases embedded in occupational or descriptive text, offering a more comprehensive representation of latent societal stereotypes.

## B.3 User Study Details

We conducted a user study using the crowdsourcing platform *Prolific*[2] to collect annotations from diverse participants. Prolific allows researchers to recruit participants according to specific demographic quotas. In our case, we applied *sex* and *ethnicity* quotas to ensure balanced participation and obtain responses from a more diverse pool of annotators.

All responses were collected using a *Google Form*[3]. Given a set of text prompts, participants were instructed to list potential biases or implicit characteristics in the form of categories and attributes. To facilitate

---

[2]https://www.prolific.com/
[3]https://docs.google.com/forms/

Table 11: Detected bias categories and corresponding attributes by different LLMs for the input prompt "A CEO".

| Category | Claude 3.7 Sonnet | GPT-4o | LLaMA 3.3 | DeepSeek V3.2 |
|---|---|---|---|---|
| Gender | male, female, non-binary | male, female, non-binary | male, female, non-binary | male, female, non-binary |
| Age | young, middle-aged, elderly | young adult, middle-aged, elderly | young adult, middle-aged, elderly | middle-aged, elderly, young adult |
| Race | White, Black, Asian, Hispanic, Middle Eastern, Indigenous | White, Asian, Black, Hispanic | White, Asian, Black, Hispanic, Middle Eastern | White, Asian, Black, Hispanic, Middle Eastern |
| Appearance / Attire | formal attire, business suit, casual business attire, professionally dressed | business suit, casual wear, formal wear | formal, business casual, luxurious, conservative, modern | well-dressed, formal attire, casual attire, professional |
| Body type / Physical ability | slim, athletic, average, plus-sized | – | – | able-bodied, uses mobility aid |
| Setting | office, boardroom, corporate headquarters, tech company, factory floor | boardroom, office, conference room | corporate, industrial, academic | – |
| Background | – | – | urban, rural | ivy league educated, self-made, inherited position, career climber |
| Pose | sitting at desk, standing, presenting, in meeting, speaking | – | – | – |
| Accessories | glasses, watch, briefcase, laptop, smartphone, tablet | – | – | – |
| Facial expression / Personality | serious, smiling, confident, authoritative, friendly | – | confident, serious, smiling, determined, approachable | authoritative, charismatic, decisive, collaborative |
| Industry / Environment | – | – | tech, finance, healthcare, manufacturing, services | technology, finance, healthcare, manufacturing, retail |
| Nationality | – | – | – | American, European, Asian, Global |

understanding, the instruction included several example answers illustrating how participants should format their responses, so that they could easily grasp our intended task. The task interface shown to participants is illustrated in Fig. 5. Furthermore, we enforced a regular-expression-based answer format to ensure consistent and well-structured annotations.

We prepared a total of 100 text prompts, consisting of 50 prompts from the Stable Bias dataset (Luccioni et al., 2023) and 50 prompts from the PartiPrompts dataset (Yu et al., 2022), as detailed in Section B.1. To maintain reasonable annotation time, these 100 prompts were divided into five separate tasks, each containing 20 prompts. Different tasks were assigned to different participants to avoid repetition.

For each task, we recruited five independent annotators, resulting in a total of 25 participants. This configuration kept the estimated completion time around 30 minutes per participant. Each participant received a compensation of **£4.5**, corresponding to an hourly rate of approximately **£9.0**, which aligns with fair payment standards on Prolific.

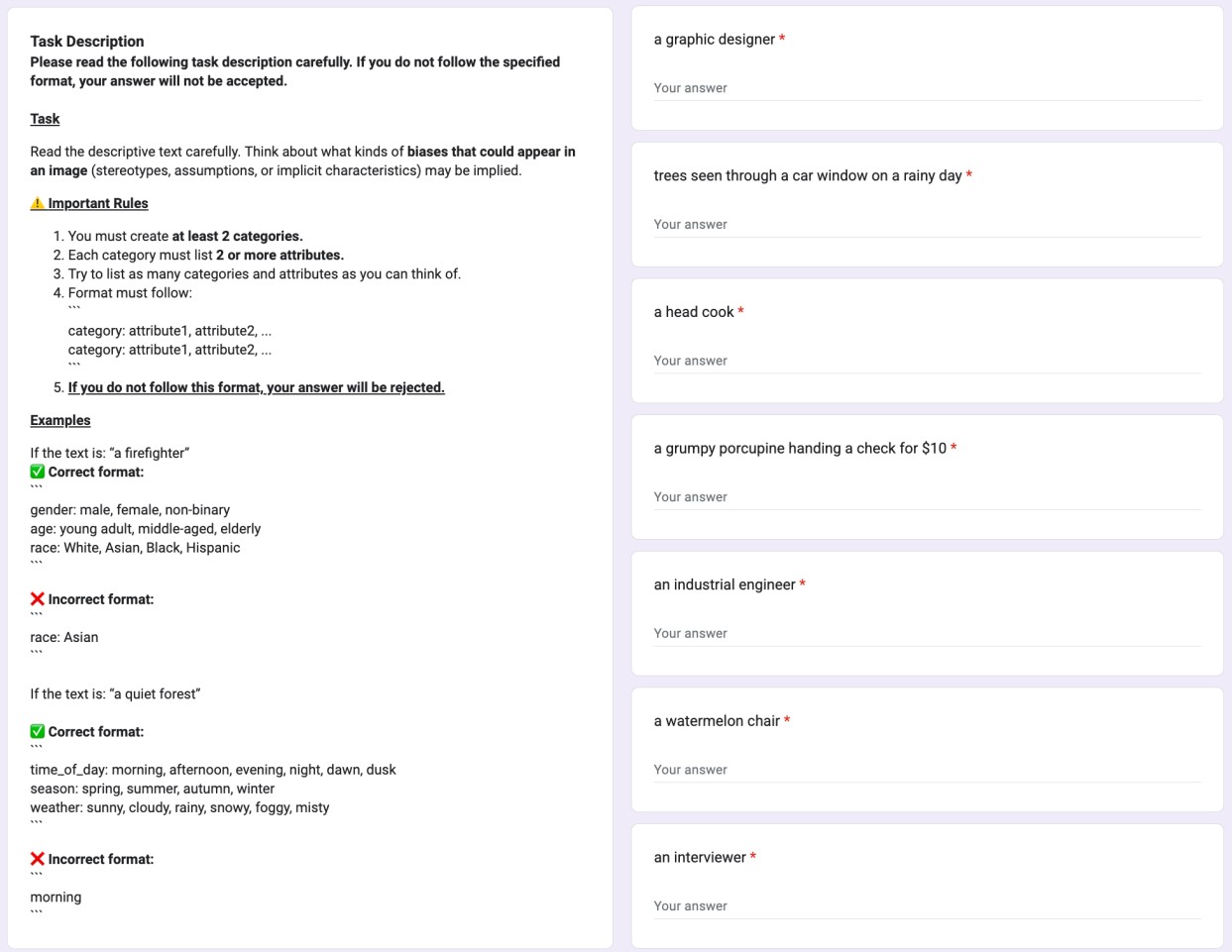

Figure 5: The task interface shown to participants, including the task description, sample prompts, and entry fields for submitting responses.

## C   Details for Societal Bias Mitigation Experiment

### C.1   Evaluation Metric Details

The FairFace classifier outputs two gender categories, *male* and *female*, and seven race categories, *White*, *Southeast Asian*, *Middle Eastern*, *Latino_Hispanic*, *Indian*, *East Asian*, and *Black*. Based on these classification results, we compute the statistical parity against the target distributions. Table 9 shows the 2024 BLS employment statistics[4] for *CEO*, *computer programmer*, *doctor*, *nurse*, and *housekeeper*. The original statistics report the percent of total employed for women, *White*, *Black or African American*, *Asian*, and *Hispanic or Latino*. To construct the BLS target distribution, we assigned the proportion of women to the probability of *female* and one minus that proportion to *male* in the gender category. For the race categories, we normalized the proportions for *White*, *Black or African American*, *Asian*, and *Hispanic or Latino* so that they sum to one[5] and used this as the target distribution. Because the FairFace classifier does not include a generic *Asian* category, we combined the probabilities of *Southeast Asian* and *East Asian* to represent *Asian* when computing statistical parity.

---

[4]https://www.bls.gov/cps/cpsaat11.htm
[5]NOTE: Estimates for the above race groups (*White*, *Black or African American*, and *Asian*) do not sum to totals because data are not presented for all races. Persons whose ethnicity is identified as *Hispanic or Latino* may be of any race.

## C.2 ENTIGEN Setup Details

As an ethical intervention to promote equitable judgment in T2I models, we added "*, if all individuals can be <PROFESSION> irrespective of their gender and race*" to the end of the input prompt.

## C.3 Fair Diffusion Setup Details

For gender editing, we predefined the prompts ["*male person*", "*female person*", "*non-binary person*"], and for race editing, the prompts ["*white person*", "*black person*", "*asian person*", "*latino person*", "*indian person*"]. During sampling, one gender prompt and one race prompt were drawn according to the target distribution, and `reverse_editing_direction=True` was used to enhance the selected attribute while suppressing the others. All editing operations employed the following hyperparameters (as in the original implementation):

- `edit_warmup_steps = 1`
- `edit_guidance_scale = 3.0`
- `edit_threshold = 0.95`
- `edit_weights = 1.0`
- `edit_momentum_scale = 0.3`
- `edit_momentum_beta = 0.6`

Except for these settings, all other parameters matched those of the original work Friedrich et al. (2023).

## C.4 FairT2I Setup Details

T2I model sampling was performed using `PNDMScheduler` with 50 sampling steps. The guidance_scale for classifier-free guidance was set to `7.5`. During inference, we set the batch size to 200 and image resolution $512 \times 512$, and a single inference on one NVIDIA H100 GPU took approximately 4 minutes.

For bias detection, we used `claude-3-7-sonnet-20250219`, and for prompt rewriting, we used `gpt-4o-mini-2024-07-18`.

The prompt used for LLM-assisted bias detection is as follows:

*Please read the following task description carefully. If you do not follow the specified format, your answer will not be accepted.*

*Task*

*Read the descriptive text carefully. Think about what kinds of biases that could appear in an image (stereotypes, assumptions, or implicit characteristics) may be implied.*

*Important Rules*

*You must create at least 2 categories.*
*Each category must list 2 or more attributes.*
*Try to list as many categories and attributes as you can think of.*
*Format must follow:*
*{*
 *"category": ["attribute1", "attribute2", ...],*
 *"category": ["attribute1", "attribute2", ...],*
*}*

*Do not include any text outside of the JSON output.*
*If you do not follow this format, your answer will be rejected.*

*Examples*
*If the text is: "a firefighter"*
*Correct format:*
*{*
  *"gender": ["male", "female", "non-binary"],*
  *"age": ["young adult", "middle-aged", "elderly"],*
  *"race": ["White", "Asian", "Black", "Hispanic"],*
*}*
*Incorrect format:*

*{*
  *"race": "Asian"*
*}*
*If the text is: "a quiet forest"*
*Correct format:*
*{*
  *"time_of_day": ["morning", "afternoon", "evening", "night", "dawn", "dusk"],*
  *"season": ["spring", "summer", "autumn", "winter"],*
  *"weather": ["sunny", "cloudy", "rainy", "snowy", "foggy", "misty"],*
*}*
*Incorrect format: morning*

The following prompt was used for the LLM-assisted fusion of the textual input **y** and the latent attribute **z**:

*Please read the following task description carefully. If you do not follow the specified format, your answer will not be accepted.*

***Task***

*Rewrite the original prompt so that it naturally integrates all specified attributes.*

***Important Rules***

- *Output must contain only the rewritten prompt text.*

- *Do not add, expand, or explain beyond the original prompt and attributes.*

- *Do not include extra descriptive words, embellishments, or commentary.*

- *Do not include explanations, metadata, or formatting outside of the rewritten prompt.*

- *The rewritten sentence must be as concise as possible, while preserving the original meaning and smoothly integrating all attributes.*

***Format***

`<rewritten prompt text>`

*If you do not follow this format, your answer will be rejected.*

***Examples***
*If the input is:*

*{*

   *Original prompt: "A portrait of a person reading a book"*
   *Attributes to include: "gender: female, age: elderly"*
*}*

***Correct format:***

*{*

   *A portrait of an elderly female person reading a book*
*}*

***Incorrect format:***

*{*

   *A detailed portrait depicting an elderly female person seated in a quiet and serene setting, holding a book gently in her hands while reading it with focused attention and calm concentration.*
*}*

### C.5 Statistical Significance Test Methodology

To determine if FairT2I achieved statistically significantly lower SP scores compared to another method (denoted as Method A), we employed a one-sided non-parametric bootstrap hypothesis test with $N_{boot} = 1000$ iterations. In each iteration $i$, demographic classifications were resampled with replacement from the set of generated images for Method A and FairT2I, maintaining the original sample sizes from which the SP scores were computed. Specifically:

- For the **uniform distribution target** (see Section C.6), overall SP scores were initially calculated from demographic classifications of all 50 occupations $\times$ 200 images/occupation $= 10,000$ images generated per method. The bootstrap procedure then resampled from these $10,000$ image classifications for each method to generate bootstrap SP scores.

- For the **BLS statistics target** (see Section C.8), SP scores were initially calculated for each occupation based on the $n = 200$ images generated per occupation per method. The bootstrap procedure then resampled from the 200 image classifications for a given occupation for each method.

SP scores were then calculated for these bootstrap samples ($\mathrm{SP}^*_{A,i}$ for Method A and $\mathrm{SP}^*_{B,i}$ for FairT2I, where B denotes FairT2I). The difference $D^*_i = \mathrm{SP}^*_{A,i} - \mathrm{SP}^*_{B,i}$ was computed for each iteration. The $p$-value was estimated as the proportion of these bootstrap differences $D^*_i$ that were less than or equal to zero: $p = \frac{\sum_{i=1}^{N_{boot}} \mathbb{I}(D^*_i \leq 0)}{N_{boot}}$, where $\mathbb{I}(\cdot)$ is the indicator function. This $p$-value tests the null hypothesis $H_0 : \mathrm{SP}_{\mathrm{FairT2I}} \geq \mathrm{SP}_{\mathrm{Method\ A}}$ against the alternative hypothesis $H_A : \mathrm{SP}_{\mathrm{FairT2I}} < \mathrm{SP}_{\mathrm{Method\ A}}$. $p < 0.05$ indicates that FairT2I is significantly better than Method A for the given SP metric.

### C.6 Detailed Results of Uniform Distribution Target

The overall SP scores for gender and race when targeting a uniform distribution are presented in Table 12. The statistical significance of FairT2I's scores compared to other methods was assessed using the bootstrap test described in Section C.5.

Table 12: Gender SP (left) and Race SP (right) scores (lower is better) from the uniform distribution. For each metric, the best score is shown in **bold**. For Original, ENTIGEN, and FairDiffusion, the $p$-value in parentheses ($p$-val) indicates if FairT2I (Ours) is significantly better than that method ($p < 0.05$ highlighted in **bold** for $\mathrm{SP_{FairT2I}} < \mathrm{SP_{Method}}$).

| Method | Gender SP ↓ | Race SP ↓ |
|---|---|---|
| Original | 0.2056 (<**0.0001**) | 0.3907 (<**0.0001**) |
| ENTIGEN | 0.0661 (<**0.0001**) | 0.2001 (<**0.0001**) |
| FairDiffusion | 0.2083 (<**0.0001**) | 0.3137 (<**0.0001**) |
| FairT2I (Ours) | **0.0123** | **0.1864** |

**Original**    **ENTIGEN**    **FairDiffusion**    **FairT2I (Ours)**

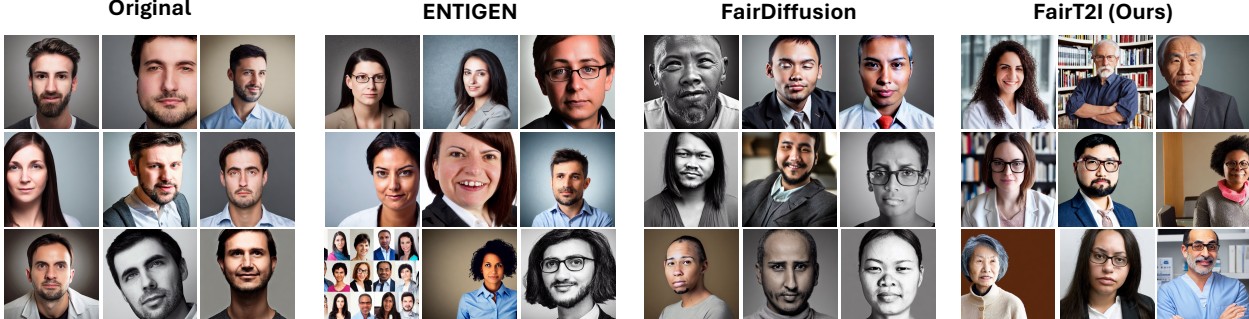

Figure 6: Generated images for the input text "*a headshot of a psychologist*" by the original Stable Diffusion, ENTIGEN, FairDiffusion, and FairT2I (Ours).

## C.7 Evaluation Results

**Discussion of uniform distribution results.** As shown in Table 12, when targeting a uniform demographic distribution across all occupations, FairT2I (Ours) achieved the best overall SP scores for both Gender SP (0.0123) and Race SP (0.1864). The statistical significance tests, detailed in Section C.5, confirm these improvements. FairT2I is shown to be significantly better than the Original method (Gender SP: 0.2056, Race SP: 0.3907), ENTIGEN (Gender SP: 0.0661, Race SP: 0.2001), and FairDiffusion (Gender SP: 0.2083, Race SP: 0.3137) for both metrics, with all associated $p$-values being less than 0.000. This indicates a strong and statistically robust improvement in fairness for overall image generation when a uniform distribution is the goal.

**Stacked Bar-Chart Visualization.** Fig. 7 presents a stacked bar-chart comparison between the target uniform distribution and the empirical distributions produced by Stable Diffusion, ENTIGEN, Fair Diffusion, and our proposed FairT2I. Among these methods, FairT2I demonstrates the closest alignment with the target distribution.

## C.8 Detailed Results of BLS Statistics Target

For the BLS statistics target, FairT2I's performance in achieving lower SP scores per occupation was also evaluated against other methods. The statistical significance of these comparisons was determined using the one-sided non-parametric bootstrap hypothesis test detailed in Section C.5. The SP scores and the corresponding $p$-values from these tests are presented in Table 13.

**Discussion of BLS statistics results** The analyses in Table 13 indicate that FairT2I generally achieves lower SP scores for both gender and race attributes compared to existing methods when targeting BLS statistics. For **Gender SP**, FairT2I consistently demonstrates statistically significant improvements ($p < 0.05$) over the Original method across all five occupations. It also shows significant advantages over FairDiffusion in most occupations (*CEO*, *computer programmer*, *nurse*, *housekeeper*). Compared to ENTIGEN, FairT2I provides significant improvements in two occupations (*CEO*, *nurse*), while for others, including a tied best performance with ENTIGEN for *housekeeper*, the differences were not statistically significant. Regarding

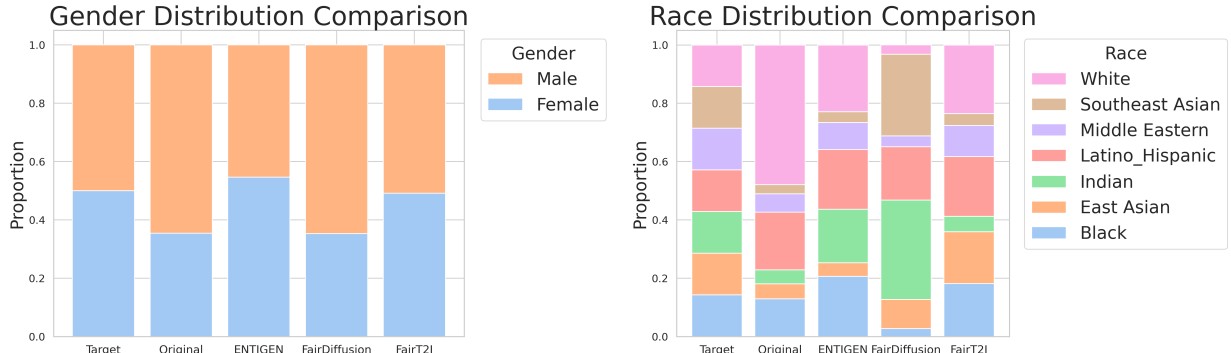

Figure 7: Comparison of the target uniform distribution and empirical distributions produced by original Stable Diffusion, ENTIGEN, FairDiffusion and FairT2I (Ours) for gender (left) and race (right), shown as stacked bar charts.

Table 13: Gender SP (top) and Race SP (bottom) scores (lower is better) when targeting BLS statistics. For each metric and occupation, the best score is shown in **bold**. For Original, ENTIGEN, and FairDiffusion, the $p$-value in parentheses ($p$-val) indicates if FairT2I (Ours) is significantly better than that method ($p < 0.05$ highlighted in **bold** for $SP_{FairT2I} < SP_{Method}$).

| | **Gender SP ↓** | | | | |
|---|---|---|---|---|---|
| **Method** | CEO | comp. prog. | doctor | nurse | housekeeper |
| Original | 0.3637 (**0.000**) | 0.1651 (**0.000**) | 0.3843 (**0.000**) | 0.1796 (**0.001**) | 0.1459 (**0.000**) |
| ENTIGEN | 0.3748 (**0.000**) | 0.0569 (0.254) | 0.1229 (0.123) | 0.1210 (**0.044**) | **0.0121** (0.547) |
| FairDiffusion | 0.1956 (**0.022**) | 0.1089 (**0.044**) | 0.1138 (0.206) | 0.1653 (**0.004**) | 0.1597 (**0.000**) |
| FairT2I (Ours) | **0.0710** | **0.0170** | **0.0550** | **0.0537** | **0.0121** |

| | **Race SP ↓** | | | | |
|---|---|---|---|---|---|
| **Method** | CEO | comp. prog. | doctor | nurse | housekeeper |
| Original | 0.6050 (**0.000**) | 0.2213 (**0.003**) | 0.2110 (**0.016**) | 0.1971 (0.234) | 1.0207 (**0.000**) |
| ENTIGEN | 0.5004 (**0.000**) | 0.6115 (**0.000**) | 0.4727 (**0.000**) | 0.4805 (**0.000**) | 0.9341 (**0.000**) |
| FairDiffusion | 0.1664 (**0.019**) | 0.1484 (**0.027**) | **0.0750** (0.752) | **0.1285** (0.714) | 0.3074 (**0.008**) |
| FairT2I (Ours) | **0.0677** | **0.0886** | 0.1137 | 0.1540 | **0.2001** |

**Race SP**, FairT2I shows robust and statistically significant improvements over ENTIGEN across all occupations. Against the Original method, FairT2I is significantly better in four out of five occupations. When compared with FairDiffusion, FairT2I offers significant gains for three occupations (*CEO*, *computer programmer*, *housekeeper*). Notably, FairDiffusion achieved lower Race SP scores for the *doctor* and *nurse* occupations, where FairT2I did not show a significant advantage.

**Stacked bar chart visualization** Fig. 8 presents stacked bar charts comparing demographic distributions of target BLS statistics and empirical distributions across five occupations. Visually, FairT2I generally demonstrates a closer alignment to the target BLS distributions for both gender and race across most occupations.

# D Details for Diversity Control Experiment

## D.1 Evaluation Metrics Details

This section provides further details on the implementation of the evaluation metrics used in Section 6 to assess image fidelity, text-image alignment, and diversity. For all trace-based diversity metrics, we generated

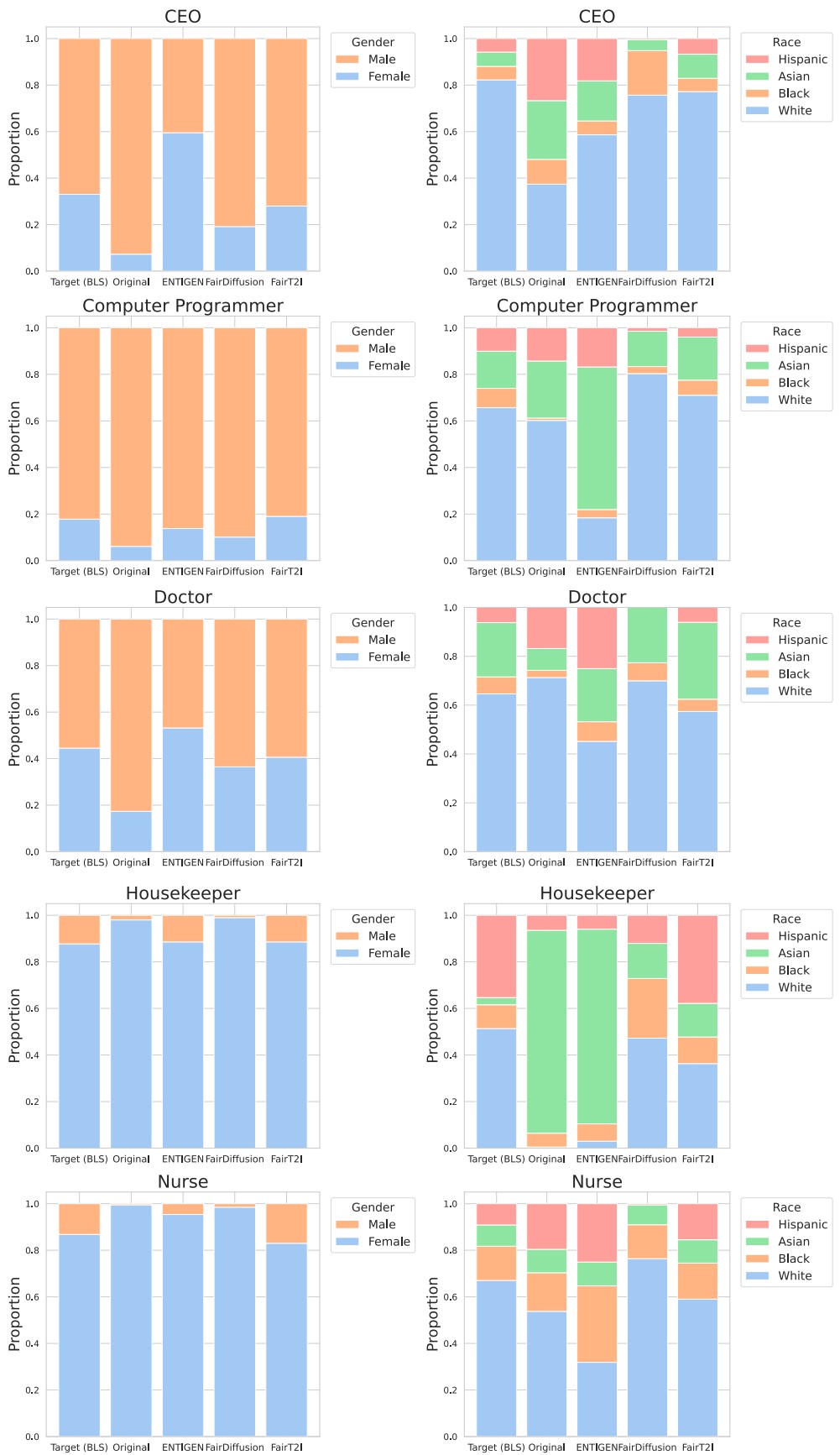

Figure 8: Comparison of the target BLS statistics and empirical distributions produced by original Stable Diffusion, ENTIGEN, FairDiffusion and FairT2I (Ours) for gender (left) and race (right), shown as stacked bar charts.

200 images for each of the 50 sampled Parti Prompts. Image features were extracted from these 200 images, and the trace of their covariance matrix was computed.

**FID.** We calculated FID Heusel et al. (2017) to assess image fidelity against the COCO Karpathy test split (Lin et al., 2014). We used the PyTorch implementation of FID from the `pytorch-fid` library[6].

**CLIPScore.** To evaluate text-image alignment, we computed CLIPScore Hessel et al. (2021).

- **Model and Library:** We utilized the OpenAI CLIP `ViT-B/32` model, accessed via the official `clip` Python package.

- **Feature Extraction:** For each (prompt, generated image) pair, the prompt text was encoded using `model.encode_text()` and the image was encoded using `model.encode_image()`. Both encoders produce L2-normalized features.

- **Score Calculation:** The CLIPScore for a single pair is the dot product of their respective normalized feature embeddings.

- **Aggregation:** The final reported CLIPScore for a given prompt is the average score across all 200 images generated for that prompt.

**CLIP Trace.** To quantify diversity using CLIP Radford et al. (2021) features, we calculated the trace of the covariance matrix of image embeddings.

- **Model and Library:** We used the OpenAI CLIP `ViT-B/32` model, via the `clip` Python package.

- **Feature Extraction:** For each of the 50 sampled prompts, we generated 200 images. All 200 images were encoded into L2-normalized feature vectors using `model.encode_image()` with a batch size of 32. This resulted in a set of 200 feature vectors per prompt.

- **Trace Calculation:** For each prompt, we computed the covariance matrix of these 200 feature vectors using `numpy.cov(features, rowvar=False)`. The CLIP Trace diversity metric is the trace of this covariance matrix, calculated using `numpy.trace()`.

**BLIP2 Trace.** Similar to CLIP Trace, we also quantified diversity using BLIP-2 Li et al. (2023) features.

- **Model and Library:** We employed the `Salesforce/blip2-opt-2.7b` model, accessed via the Hugging Face `transformers` library, specifically using `Blip2Processor` for preprocessing and `Blip2Model` for feature extraction.

- **Feature Extraction:** For each prompt, the 200 generated images were processed. Image features were obtained from the `pooler_output` of `model.get_image_features()`.

- **Trace Calculation:** Analogous to CLIP Trace, for each prompt, we computed the covariance matrix of the 200 BLIP-2 image feature vectors and then calculated its trace.

## D.2 FairT2I Setup Details

T2I model sampling was performed using `FlowMatchEulerDiscreteScheduler` with 28 sampling steps. The guidance_scale for classifier-free guidance was set to 4.0. During inference, we set the batch size to 200 and image resolution $1024 \times 1024$, and a single inference on one NVIDIA H100 GPU took approximately 4 minutes. For bias detection, we used `claude-3-7-sonnet-20250219`, and for prompt rewriting, we used `gpt-4o-mini-2024-07-18`.

The same prompt in Section C.4 was used for LLM-assisted bias detection is as follows. We utilized the same prompt for integration of the input text **y** and the sensitive attribute **z** with Section C.4.

---

[6]`https://github.com/mseitzer/pytorch-fid`

### D.3 Statistical Significance Test Methodology

To assess the statistical significance of the differences observed in diversity (BLIP2 Trace, CLIP Trace) and text-image alignment (CLIP Score) metrics (see Section 6 in the main paper), we performed one-sided Mann-Whitney U tests. This non-parametric test was chosen as it does not assume a specific distribution (e.g., normality) for the data and is suitable for comparing two independent samples of per-prompt scores. For each comparison between two methods (Group 1 and Group 2, as detailed in Table 14), we tested the alternative hypothesis ($H_A$) that the scores from Group 1 are stochastically greater than those from Group 2. The detailed results of these tests, including $p$-values and findings, are presented in Table 14. Note that statistical tests for FID scores were not performed due to the high computational cost associated with generating multiple full sets of images for each condition required for robust FID variance estimation. FID scores are reported as single values based on one comprehensive generation run per method.

### D.4 Discussion of Diversity, Alignment, and Fidelity Results

The statistical analyses presented in Table 14, along with reported FID scores, provide insights into FairT2I's performance in terms of image diversity, text-image alignment, and image fidelity, relative to baseline Stable Diffusion models with varying classifier-free guidance (CFG) scales.

**Diversity (Trace Scores).** FairT2I (itself using a CFG scale of 4.0) demonstrates a statistically significant increase in diversity, as measured by both BLIP2 Trace and CLIP Trace, when compared to the original Stable Diffusion model using higher or equivalent CFG scales (i.e., CFG 7.0 and CFG 4.0). Specifically, for both trace metrics, FairT2I's scores were significantly higher (all $p < 0.00001$, except for BLIP2 Trace vs. Original CFG 4.0 where $p = 0.000002$). When compared to the original model with a very low CFG scale (CFG 1.0), which typically yields high diversity, the results are nuanced. While the original CFG 1.0 model achieved numerically higher mean trace scores for both BLIP2 Trace and CLIP Trace compared to FairT2I, these differences were not found to be statistically significant (BLIP2 Trace $p = 0.255$; CLIP Trace $p = 0.069$ when testing $H_A$: Original CFG 1.0 > FairT2I). This suggests FairT2I achieves a level of diversity that is statistically comparable to that of the highly diverse CFG 1.0 setting.

**Text-image alignment (CLIP Score).** For CLIP Scores, the original model with CFG scales of 7.0 and 4.0 yielded statistically significantly higher scores than FairT2I ($p = 0.021$ and $p = 0.008$, respectively, when testing $H_A$: Original > FairT2I). This indicates a slight, yet statistically significant, decrease in text-image alignment for FairT2I when compared to these higher CFG scale baselines. However, when comparing FairT2I to the original model with CFG 1.0, there was no statistically significant evidence that Original CFG 1.0's CLIP Score was higher than FairT2I's ($p = 0.664$). Numerically, FairT2I's mean CLIP Score was slightly higher than that of Original CFG 1.0. This suggests a trade-off: increasing diversity via FairT2I might involve a small compromise in text-image alignment relative to high-guidance settings, but the alignment remains competitive, especially when compared to other high-diversity configurations like CFG 1.0.

**Image Fidelity (FID) and overall balance.** While statistical tests were not performed for FID due to computational expense, the reported scores (Original CFG 7.0: 27.13, Original CFG 4.0: 26.46, Original CFG 1.0: 34.47, FairT2I: 26.24) indicate that FairT2I achieves the best image fidelity (lowest FID score). It surpasses or matches the fidelity of the original model at CFG 4.0 and 7.0, and is substantially better than the CFG 1.0 setting, which shows a marked degradation in fidelity.

**Conclusion on trade-offs.** Taken together, FairT2I appears to offer a favorable balance. It significantly enhances image diversity over standard CFG settings (4.0 and 7.0) and achieves diversity levels statistically comparable to the very low CFG 1.0 setting. While there is a statistically significant, albeit modest, reduction in CLIPScore compared to using CFG 4.0 and 7.0 with the original model, FairT2I maintains excellent image fidelity (best FID). This presents a notable advantage over simply lowering the CFG scale to 1.0 with the original model, which boosts diversity but at a considerable cost to image fidelity and without a clear advantage in text-image alignment over FairT2I. Therefore, FairT2I effectively increases diversity while largely preserving image quality and maintaining a competitive level of text-image alignment.

Table 14: Statistical comparison of FairT2I with Original Stable Diffusion (various CFG scales) using one-sided Mann-Whitney U tests. For all comparisons, the alternative hypothesis ($H_A$) tested is that the median score of **Group 1** is stochastically greater than that of **Group 2**. Significance level $\alpha = 0.05$.

| Metric | Group 1 | Group 2 | $p$-value | Finding |
|---|---|---|---|---|
| | FairT2I | CFG 7.0 | <0.000001 | Group 1 significantly higher |
| BLIP2 Trace | FairT2I | CFG 4.0 | 0.000002 | Group 1 significantly higher |
| | CFG 1.0 | CFG 4.0 | 0.255153 | Group 1 not significantly higher |
| | FairT2I | CFG 7.0 | <0.000001 | Group 1 significantly higher |
| CLIP Trace | FairT2I | CFG 4.0 | <0.000001 | Group 1 significantly higher |
| | CFG 1.0 | CFG 4.0 | 0.068690 | Group 1 not significantly higher |
| | CFG 7.0 | FairT2I | 0.020644 | Group 1 significantly higher |
| CLIP Score | CFG 4.0 | FairT2I | 0.008392 | Group 1 significantly higher |
| | CFG 1.0 | FairT2I | 0.664210 | Group 1 not significantly higher |

## D.5 User Study Details

We conducted a user study using the crowdsourcing platform *Prolific* to collect annotations from a diverse group of participants. Prolific allows researchers to recruit participants according to predefined demographic quotas; in our case, we applied *sex* and *ethnicity* quotas to ensure balanced participation and obtain responses from a demographically varied pool of annotators.

All responses were collected via a *Google Form*. For each task, participants were presented with a text prompt and sets of images generated by four anonymized models (*Model A*, *Model B*, *Model C*, and *Model D*). They were asked to rate each model on a five-point Likert scale (1 = worst, 5 = best) along three criteria: *diversity*, *image quality*, and *text–image alignment*. Each model output was displayed as a $3 \times 3$ image grid aligned horizontally with white padding to facilitate visual comparison.

To promote consistent and reliable responses, the instruction section included concise explanations and rubrics defining each evaluation criterion. The complete task interface shown to participants is illustrated in Fig. 9. Additionally, we enforced a regular-expression-based input format to ensure well-structured and standardized annotations across all responses.

We sampled 50 prompts from the Parti Prompts dataset (Yu et al., 2022), as detailed in Section B.1. To keep the annotation workload manageable, these 50 prompts were divided into five separate tasks, each containing 10 prompts. Each task was assigned to a distinct group of participants to prevent repetition across annotators.

For each task, we recruited 20 independent annotators, resulting in a total of 100 participants overall. This configuration maintained an average completion time of approximately 20 minutes per participant. Each participant received an hourly rate of approximately **£6.0**, which is slightly below Prolific's recommended fair payment rate but was deemed reasonable given the short task duration and minimal cognitive load.

## D.6 Visual Results

This section presents additional visual comparisons for several input text prompts in Parti Prompt dataset. Figures 10–13 illustrate images generated using our proposed FairT2I method alongside those generated with classifier-free guidance (CFG) at varying scales (7.0, 4.0, and 1.0). As noted in the captions, a guidance scale of 1.0 effectively corresponds to generation without the influence of classifier-free guidance. Our FairT2I method demonstrates an enhanced capability to diversify generation results, particularly for prompts that do not necessarily involve humans—spanning categories such as animals (Figures 10 and 13), objects (Figure 11), and abstract concepts (Figure 12). Each figure allows for a qualitative assessment of the different generation conditions for its respective prompt.

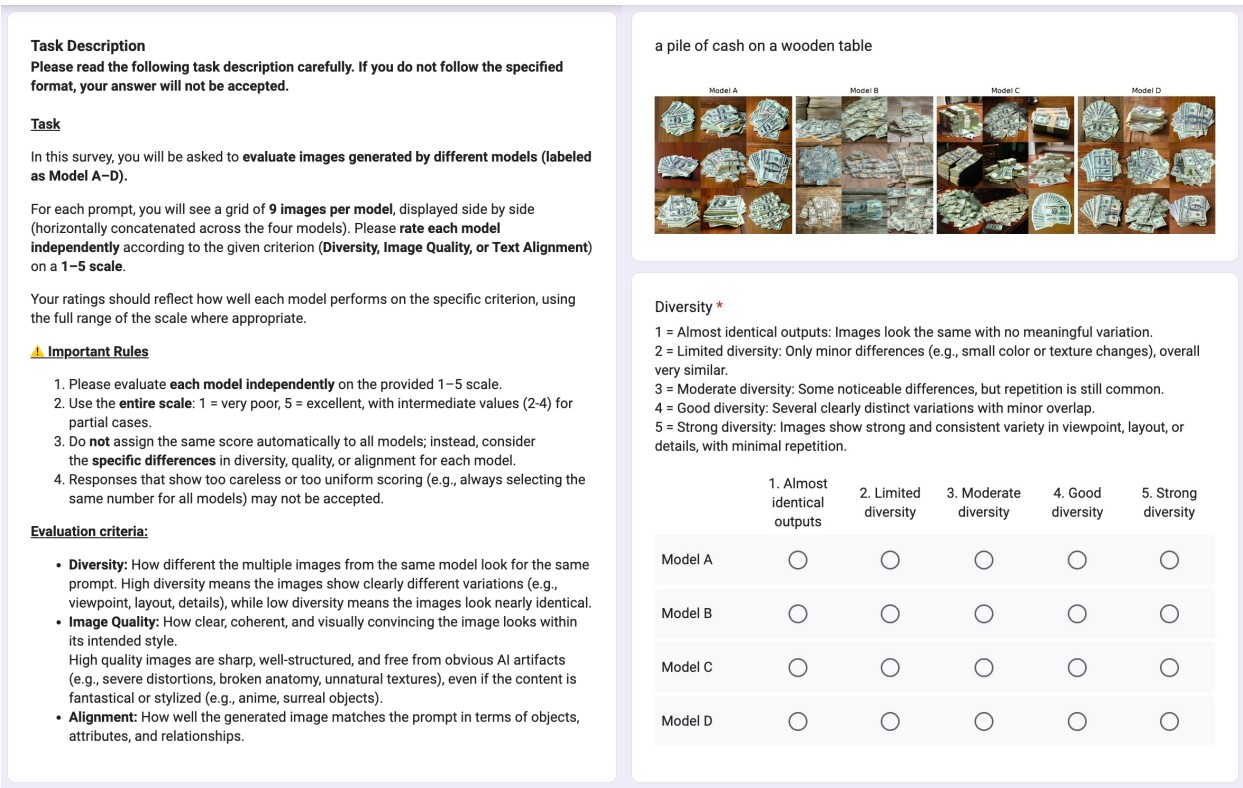

Figure 9: User study interface provided to participants, including the task description, input prompts, generated images, and the multi-grid answer form for annotators.

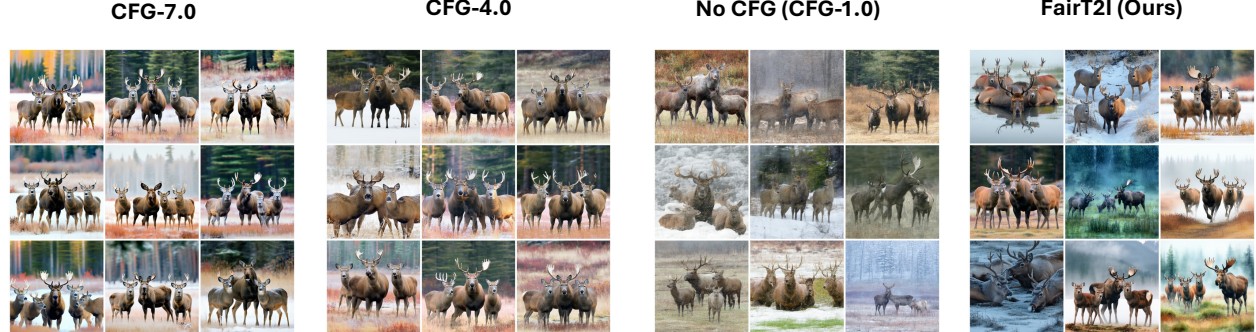

Figure 10: Generated images for the input text "*Four deer surrounding a moose.*" by classifier-free guidance (CFG) at guidance scales 7.0, 4.0, and 1.0 and FairT2I (Ours). A guidance scale of 1.0 corresponds to generation without CFG.

**CFG-7.0**  **CFG-4.0**  **No CFG (CFG-1.0)**  **FairT2I (Ours)**

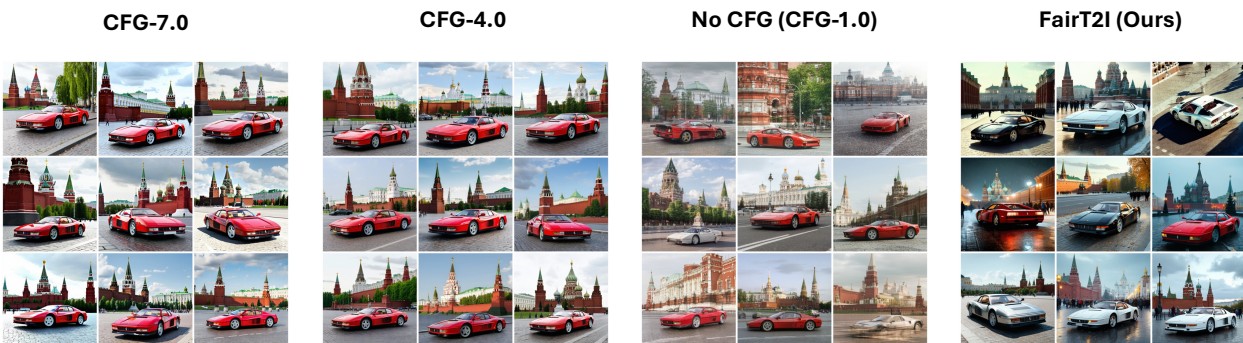

Figure 11: Generated images for the input text "*a Ferrari Testarossa in front of the Kremlin*" by classifier-free guidance (CFG) at guidance scales 7.0, 4.0, and 1.0 and FairT2I (Ours). A guidance scale of 1.0 corresponds to generation without CFG.

**CFG-7.0**  **CFG-4.0**  **No CFG (CFG-1.0)**  **FairT2I (Ours)**

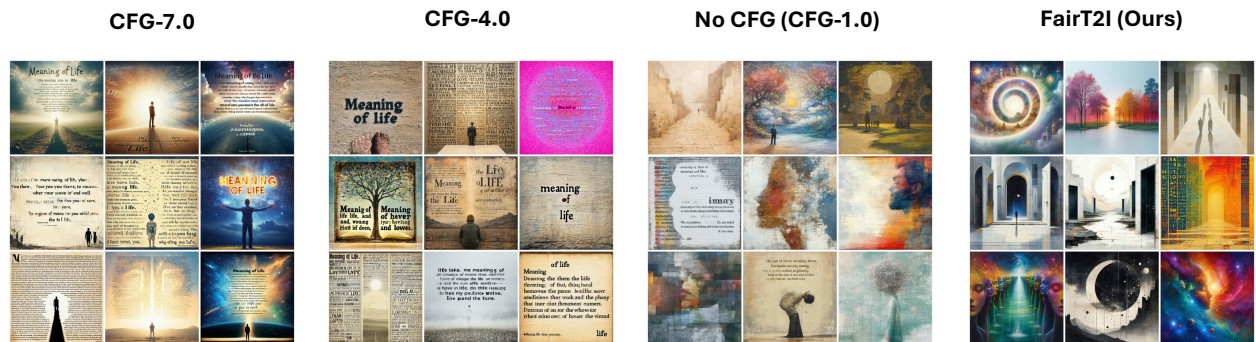

Figure 12: Generated images for the input text "*meaning of life*" by classifier-free guidance (CFG) at guidance scales 7.0, 4.0, and 1.0 and FairT2I (Ours). A guidance scale of 1.0 corresponds to generation without CFG.

**CFG-7.0**  **CFG-4.0**  **No CFG (CFG-1.0)**  **FairT2I (Ours)**

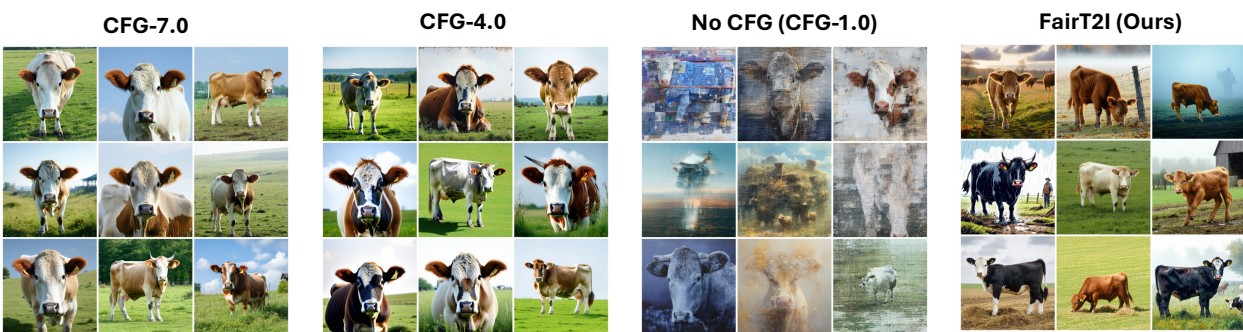

Figure 13: Generated images for the input text "*a cow*" by classifier-free guidance (CFG) at guidance scales 7.0, 4.0, and 1.0 and FairT2I (Ours). A guidance scale of 1.0 corresponds to generation without CFG.

