# OpenReview forum: "FairT2I: Latent Variable Guidance for Training-Free Bias Mitigation with LLM-Assisted Bias Detection"
_TMLR — Decision pending for TMLR_

### Review · Reviewer_PZmy · 2026-04-22

**Summary Of Contributions:**

The paper introduces FairT2I, a training-free inference-time method for reducing bias in text-to-image generation. The method uses an LLM to identify bias-relevant latent attributes in a prompt, resamples those attributes, and then guides generation using an attribute-conditioned latent guidance formulation. The paper evaluates the method on occupational bias benchmarks and on Parti Prompts, arguing improved demographic fairness and a better diversity-quality trade-off than prior inference-time baselines.

However, I would like to note that I am not entirely familiar with this field (text2image generation).

Strength:
   * The problem addressed (bias mitigation) is important and practical.
   * The method is appealing because it operates entirely at inference time.
   * The overall pipeline is easy to understand.

Weakness:
   * The contribution of each component is not cleanly isolated. Since the framework combines multiple components, an ablation study separating the effect of each component is necessary.
   * Some empirical claims are slightly overstated. For example, the paper claims improved diversity without sacrificing prompt fidelity, but the CLIPScore is lower than for CFG-4.0 and CFG-7.0.
   * I do not find the LLM bias-detection evaluation fully convincing. Much of the comparison focuses on the number of categories and attributes detected, rather than on their correctness, precision, or downstream usefulness.
   * Proposition 1’s decomposition appears to require replacing the posterior weight p(z∣x,y) with the prior p(z∣y). In Appendix A, this is justified via an assumption of conditional independence between x and z given y, but that same assumption would also imply p(x∣z,y)=p(x∣y), making the latent attribute z irrelevant to generation. Thus, the derivation is either not exact in the nontrivial case, or becomes vacuous in the case where the assumption holds.

**Audience:**

Yes

**Audience Explanation:**

I think this paper would interest readers working on fair generative modeling.

**Claims And Evidence:**

Yes

**Claims Explanation:**

I think the claims are partially supported but not yet fully convincing. Please refer to the weaknesses.

**Requested Changes:**

Please refer to the weaknesses section. I do think a clear ablation study is necessary.

---

> ### Author Response · Authors · 2026-05-05
>
> ### Concern 1: The contribution of each component is not cleanly isolated.
>
> We thank the reviewer for raising the issue of component-level isolation. FairT2I consists of three main components: LLM-assisted bias detection, attribute resampling, and latent variable guidance. We discuss each component separately below.
>
> For LLM-assisted bias detection, a seemingly direct ablation would be to compare against a debiasing method based on a fixed predefined bias table. However, in our setting, this comparison is not fully informative. Attribute resampling requires sampling attributes from an attribute table according to a target fair distribution. If the target distribution is already defined over known attributes, such as the FairFace race labels, namely White, Black, Indian, East Asian, Southeast Asian, Middle Eastern, and Latino, then a fixed-bias baseline can be given exactly these target attributes, sample uniformly from them, construct augmented prompts, and generate images. Such a baseline is effectively provided with the target attribute space in advance and can naturally produce an image distribution close to the target uniform distribution.
>
> Therefore, the central question for LLM-assisted bias detection is not whether a hand-specified attribute table can work when the target labels are already known, but whether the model can construct appropriate bias categories and bias attributes from open-ended prompts without relying on a predefined table. This is precisely what we evaluate in Section 4, where we compare LLM-detected categories and attributes with human annotations. We also compare different LLMs in Appendix B.
>
> For attribute resampling, Section 5.1 evaluates two different target distributions: a uniform target and BLS employment statistics. This comparison tests whether the proposed resampling mechanism remains effective when the target distribution changes from an intuitive fairness target to a real-world demographic prior. Thus, this experiment directly examines the role of the attribute resampling target in FairT2I.
>
> For latent variable guidance, the current manuscript does not yet isolate the approximation clearly enough. Our implementation uses a Monte Carlo approximation with sample size $N=1$. We will add an experiment varying the number of Monte Carlo samples and report both inference time and bias mitigation performance. This will quantify the trade-off between computational cost and the accuracy of the latent-guidance approximation.
>
> ---
>
> ### Concern 2: Some empirical claims are slightly overstated.
>
> We thank the reviewer for pointing out this wording issue. The phrase “without sacrificing prompt fidelity” is stronger than what the CLIPScore result alone supports, because Table 5 shows that FairT2I has a lower CLIPScore than CFG-4.0 and CFG-7.0.
>
> At the same time, CLIPScore is an automatic proxy for image-text compatibility and does not fully capture human-perceived prompt fidelity in a fairness-sensitive setting [1]. Prior work has also shown that CLIP-based representations can encode demographic and stereotypical biases across attributes such as gender, race or ethnicity, and age [2,3]. Therefore, images with more diverse or societally fair attribute distributions may receive lower CLIPScore when they deviate from demographic or stereotypical associations encoded in CLIP. For this reason, we also evaluate text-image alignment through human judgment.
>
> In Section 6.3, Table 7 reports mean alignment scores of 3.152 for CFG-1.0, 3.833 for CFG-4.0, 3.862 for CFG-7.0, and 3.682 for FairT2I. Table 8 further shows that FairT2I significantly outperforms CFG-1.0 in alignment, while the differences from CFG-4.0 and CFG-7.0 are not statistically significant under the tests reported in the current draft.
>
> We will revise the wording of the claim. Instead of stating that FairT2I improves diversity “without sacrificing prompt fidelity,” we will state that FairT2I improves diversity while maintaining competitive human-rated prompt-image alignment and image quality.
>
> ### References
>
> [1] Jack Hessel, Ari Holtzman, Maxwell Forbes, Ronan Le Bras, and Yejin Choi. CLIPScore: A Reference-free Evaluation Metric for Image Captioning. EMNLP 2021.
>
> [2] Kimia Hamidieh, Haoran Zhang, Walter Gerych, Thomas Hartvigsen, and Marzyeh Ghassemi. Identifying Implicit Social Biases in Vision-Language Models. AIES 2024.
>
> [3] Vatsal Baherwani and Joseph Vincent. Racial and Gender Stereotypes Encoded Into CLIP Representations. ICLR 2024 Tiny Papers.

---

> > ### Author Response · Authors · 2026-05-05
> >
> > ### Concern 3: The LLM bias-detection evaluation is not fully convincing.
> >
> > We clarify that the goal of Section 4 is not to establish a unique ground-truth bias table for each prompt. The goal is to evaluate whether LLMs can construct candidate bias tables that are useful for FairT2I, namely tables with broad category coverage and stable attribute construction.
> >
> > For open-ended social-bias annotation, many prompts admit multiple plausible category-attribute sets. A precision-only evaluation against a single fixed reference table would penalize reasonable attributes that are absent from that reference, even when they are useful for downstream controlled generation. This is why we compare the LLM outputs with both the mean individual human response and the union of multiple human annotations.
> >
> > In our annotation study, we used the same 100 prompts as in the evaluation datasets: 50 prompts from Stable Bias and 50 prompts from PartiPrompts. These prompts were divided into five tasks of 20 prompts each, and each task was assigned to five independent annotators, giving 25 participants in total. Annotators listed possible bias categories and multiple attributes per category. The same few-shot examples were used in the human instructions and in the LLM system prompt.
> >
> > Figure 3 reports Human Mean, Human Union, and LLM. The human union is included because individual human annotations show high variability, and the union over multiple annotators provides a stronger human reference than the mean individual response alone. Table 1 further shows that the LLM does not merely copy the few-shot examples. For race attributes in the Stable Bias subset, humans most frequently reproduce “White, Asian, Black, Hispanic,” while the LLM more systematically expands the set to include attributes such as Middle Eastern and Indigenous.
> >
> > These results support a narrower claim: LLMs provide a scalable and comparatively consistent mechanism for constructing candidate bias tables for downstream bias-aware generation. We will clarify this evaluation objective in the manuscript, while keeping the evaluation protocol unchanged.
> >
> > ---
> >
> > ### Concern 4: Proposition 1 appears to replace the posterior weight with the prior weight.
> >
> > We thank the reviewer for identifying that the current statement of Proposition 1 is too strong. The exact latent score decomposition should use the posterior weight $p(z\mid x_t,y)$, not the prior weight $p(z\mid y)$.
> >
> > We will revise the method section accordingly. The revised presentation will first state the exact posterior-weighted decomposition:
> >
> > $$\nabla_{x_t}\log p_t(x_t\mid y)=\sum_z p(z\mid x_t,y)\nabla_{x_t}\log p_t(x_t\mid z,y).$$
> >
> > We will then present the practical approximation used by FairT2I:
> >
> > $$\nabla_{x_t}\log p_t(x_t\mid y)\approx\sum_z p(z\mid y)\nabla_{x_t}\log p_t(x_t\mid z,y).$$
> >
> > To support this approximation, we add a high-noise prior approximation lemma. Under the Gaussian forward process $X_t=a_tX_0+\sigma_t\epsilon$ with $\epsilon\sim\mathcal{N}(0,I)$, the conditional Markov structure $Z\to X_0\to X_t$ given $Y=y$, and $\Sigma_y=\mathrm{Cov}(X_0\mid Y=y)$, the lemma bounds the expected total variation distance as
> >
> > $$\mathbb{E}_{x_t\sim p_t(\cdot\mid y)}[\mathrm{TV}(p(z\mid x_t,y),p(z\mid y))]\le\frac{|a_t|}{2\sigma_t}\sqrt{\mathrm{tr}(\Sigma_y)}.$$
> >
> > Thus, when the diffusion SNR $a_t^2/\sigma_t^2$ is small, $p(z\mid x_t,y)$ is close to $p(z\mid y)$ on average. This does not make the approximation exact over the full denoising trajectory, but it provides theoretical support for the approximation in the high-noise regime.
> >
> > The implemented method and empirical results remain unchanged.

---

### Review · Reviewer_ojkx · 2026-05-13

**Summary Of Contributions:**

- The authors propose FairT2I, a training-free framework that uses an LLM to dynamically detect implicit bias categories in prompts and applies the latent variable guidance rule to re-weight attribute-conditioned components during image generation.
- Intensive testing on several benchmark datasets demonstrates that FairT2I achieves some interesting trade-off between fairness and generation quality.

**Audience:**

Yes

**Audience Explanation:**

This paper extends the standard CFG methods with a three-stage method for bias mitigation on text-to-image generation. This work may interests researchers working on text-to-image generation, as well as researchers who are looking for simple bias mitigation methods.

**Claims And Evidence:**

Yes

**Claims Explanation:**

- FairT2I  balances image outputs to match target distributions,  achieving lower statistical parity scores than baseline methods.
- In a human evaluation with 100 participants, FairT2I earned the highest mean scores for both diversity and image quality, showing statistically significant improvements over standard classifier-free guidance.

**Requested Changes:**

- I have concerns regarding the derivation of the first equation in Section 3.1. Specifically, it is not immediately clear how this formulation aligns with the standard framework established by Ho & Salimans (2022). To ensure technical rigor, I recommend including a more explicit step-by-step derivation (e.g., in the Appendix) to bridge this connection for the reader.
- Regarding the final equation in Section 3.2, the claim that this represents an unbiased estimate of the true score requires further theoretical justification. Furthermore, when the latent attribute space $\mathcal{Z}$ is high-dimensional or large, a single Monte Carlo sample may not provide sufficient coverage. It would be beneficial to include a sensitivity analysis or empirical comparison demonstrating the trade-offs between different sample counts and estimation accuracy.
- The empirical results currently lack sufficient consistency to fully support the paper's claims. Specifically, in several key dimensions, the proposed method is outperformed by existing baselines, such as Alignment in human evaluation. This may implies the model may fail to follow the instruction when trying to generate unbiased outputs.

---

> ### Author Response · Authors · 2026-06-19
>
> We thank the reviewer for carefully reading the manuscript and for raising these helpful concerns. We have revised the manuscript to clarify the theoretical derivation, the role of the Monte Carlo estimator, and the empirical interpretation of the human evaluation results.
>
> ### Concern 1: Derivation of the first equation in Section 3.1.
>
> The equation is intended to express the standard classifier-free guidance score interpolation,
>
> $$
> (1-\omega)\nabla_{\mathbf{x}}\log p(\mathbf{x})
> +
> \omega\nabla_{\mathbf{x}}\log p(\mathbf{x}\mid\mathbf{y}),
> $$
>
> which interpolates between the unconditional score and the conditional score.
>
> In the revised manuscript, we replace the notation $p_w(\mathbf{x},\mathbf{y})$ with $p_\omega(\mathbf{x}\mid\mathbf{y})$, use $\omega$ consistently for the guidance scale, and add a clearer transition to Section 3.2. The added explanation clarifies that FairT2I extends this CFG view by introducing a latent attribute variable $\mathbf{z}$ and combining attribute-conditioned components according to a target attribute distribution.

---

> > ### Author Response · Authors · 2026-06-19
> >
> > ### Concern 2: Empirical consistency and alignment in human evaluation.
> >
> > FairT2I is designed to improve the trade-off among demographic fairness, diversity, image quality, and prompt-image alignment, rather than to maximize a single metric in isolation. Stronger CFG scales can yield higher alignment scores, but they may also preserve the model's original demographic tendencies. Therefore, the empirical claim should be interpreted as a fairness-diversity-quality-alignment trade-off.
> >
> > Regarding the reviewer’s concern about instruction following, the human evaluation does not indicate that FairT2I fails to follow the prompt. As reported in Table 8, the alignment difference between FairT2I and the stronger CFG baselines is not statistically significant under the reported tests, while FairT2I significantly improves alignment over CFG-1.0. Thus, the results support the interpretation that FairT2I maintains competitive human-rated alignment while improving demographic fairness and diversity.
> >
> > We have revised the manuscript to avoid overstating the alignment claim. Instead of stating that FairT2I improves diversity “without sacrificing prompt fidelity,” the revised manuscript states that FairT2I improves diversity while maintaining competitive human-rated prompt-image alignment and image quality.

---

### Review · Reviewer_Lxo8 · 2026-06-05

**Summary Of Contributions:**

1. This paper aims in addressing the problem of societal biases amplified by pre-training in the large amout of the web data. This paper proposes the FairT2I method: a training-free framework leveraging LLM for detecting and mitigating this bias during the generation.

2. This FairT2I has three components/steps: 1. Detecting the bias through prompting the LLM. 2. attribute resampling generates bias-aware prompts. 3. Latent variable guidance / bias-aware generation.

3. The performance shown in the table shows that the FairT2I can perform the success in mitigating the society bias. At the same time, the diversity of generation from the model shows its advancement.

**Audience:**

Yes

**Audience Explanation:**

Neat method.

**Broader Impact Concerns:**

No ethical

**Claims And Evidence:**

Yes

**Claims Explanation:**

The performance shown in the table shows the advancement of the method.

**Requested Changes:**

1. The whole pipeline is sampling attribute, rewriting the prompts and T2I generation. The method is neat while it dose not show the advacement.

2. However, without the training, the whole implementation of diversity and mitigation of bias raise from the edition of the prompt. However, would this bias also exist in the prompts generated from the LLM?

3. At the same time, LLM can also have the bias. How would you mitigate the bias from LLM? Would the attributes sampling can also have the bias from the large web data?

---

> ### Author Response · Authors · 2026-06-19
>
> We thank the reviewer for the positive assessment of our method and for raising important questions about the source of the improvement and the role of the LLM in FairT2I.
>
> ### Concern 1: Advancement beyond attribute sampling, prompt rewriting, and T2I generation.
> We would like to clarify that FairT2I is not merely a pipeline of prompt rewriting followed by standard text-to-image generation. Its advancement over existing inference-time bias mitigation methods lies in the combination of the following three properties.
> First, FairT2I is training-free. It does not require retraining or fine-tuning the text-to-image model for bias mitigation. This is important because new text-to-image models are continuously released, often with different text encoders and different generation backbones. A method that requires model-specific training must be re-applied whenever a new backbone is introduced. In contrast, FairT2I operates at inference time and is therefore compatible with different text encoders and T2I backbones. It can also benefit from future improvements in LLMs used for bias-category and bias-attribute construction.
> Second, FairT2I is prompt-adaptive and is not restricted to a fixed, manually predefined bias taxonomy. The bias categories and attributes relevant to a prompt are not universal; they can depend on the prompt, cultural context, and time period. Existing methods typically require the designer to specify in advance which bias attributes should be considered and often assume a fixed target fair distribution. FairT2I separates attribute discovery from target distribution specification: the LLM constructs candidate bias categories and attributes from the given prompt, while attribute resampling can use a user-specified or externally specified target distribution. Section 5.1 demonstrates this flexibility by evaluating two different target distributions, namely a uniform distribution and BLS employment statistics.
> Third, FairT2I provides a latent-variable formulation for attribute resampling and guidance. Prior methods such as Fair Diffusion and ITI-GEN also use mechanisms related to resampling attributes from a fair distribution, but they primarily instantiate this idea through prompt-level procedures and mainly consider a uniform target distribution. In contrast, FairT2I formulates bias-aware generation through latent attributes and derives latent variable guidance for combining attribute-conditioned components. In the revised manuscript, Section 3.2 and Appendix A.2 further clarify the theoretical motivation behind attribute resampling and latent variable guidance.
> Therefore, the contribution is not simply that FairT2I rewrites prompts. Rather, FairT2I provides a general, training-free, prompt-adaptive, and theoretically motivated framework for bias-aware text-to-image generation.

---

> > ### Author Response · Authors · 2026-06-19
> >
> > ### Concern 2: Bias in LLM-generated prompts and attribute sampling.
> > We would like to clarify that FairT2I does not rely on unconstrained free-form prompt rewriting by the LLM. The LLM output is structured into explicit bias categories and attributes, and the generated prompts are used as attribute-conditioned components rather than as the sole mechanism for debiasing. The sampled attributes are controlled by attribute resampling, so the generated prompts are not simply reflections of the LLM's implicit demographic associations.
> > It is also important to distinguish the construction of the candidate attribute space from the choice of the sampling distribution. FairT2I uses the LLM to propose candidate bias categories and attributes for an open-ended prompt. In contrast, the target distribution used for attribute resampling is specified separately, for example as a uniform distribution, BLS employment statistics, or a user-specified distribution. Therefore, FairT2I does not sample attributes according to the demographic frequencies implicitly represented by the LLM or inherited from large-scale web data. This separation between attribute discovery and target distribution specification is a key design choice of FairT2I.
> > We also evaluate the quality and coverage of the LLM-generated bias categories and attributes. Figure 3 reports Human Mean, Human Union, and LLM results. The human union is included because individual human annotations show high variability, and the union over multiple annotators provides a stronger human reference than the mean individual response alone. Table 1 further shows that the LLM does not merely copy the few-shot examples. For race attributes in the Stable Bias subset, humans most frequently reproduce “White, Asian, Black, Hispanic,” while the LLM more systematically expands the set to include attributes such as Middle Eastern and Indigenous.
> > These results suggest that the LLM-generated attributes are not merely reproducing a narrow set of stereotypical labels. Rather, the LLM provides a scalable mechanism for constructing candidate bias attributes with broad coverage, while the target distribution in attribute resampling controls how these attributes are sampled. Moreover, FairT2I is compatible with human-curated or user-specified attribute tables. Thus, in applications requiring a specific fairness policy or domain-specific taxonomy, the LLM-generated table can be edited or replaced while keeping the same attribute resampling and latent variable guidance framework.